# Linkage between endosomal escape of LNP-mRNA and loading into EVs for transport to other cells

Marco Maugeri [1], Muhammad Nawaz [1], Alexandros Papadimitriou[1], Annelie Angerfors [2], Alessandro Camponeschi[1], Manli Na[1], Mikko Hölttä[3], Pia Skantze[2], Svante Johansson[2], Martina Sundqvist [1], Johnny Lindquist[3], Tomas Kjellman[2], Inga-Lill Mårtensson[1], Tao Jin [1], Per Sunnerhagen [4], Sofia Östman[5], Lennart Lindfors[2] & Hadi Valadi [1]

RNA-based therapeutics hold great promise for treating diseases and lipid nanoparticles (LNPs) represent the most advanced platform for RNA delivery. However, the fate of the LNP-mRNA after endosome-engulfing and escape from the autophagy-lysosomal pathway remains unclear. To investigate this, mRNA (encoding human erythropoietin) was delivered to cells using LNPs, which shows, for the first time, a link between LNP-mRNA endocytosis and its packaging into extracellular vesicles (endo-EVs: secreted after the endocytosis of LNP-mRNA). Endosomal escape of LNP-mRNA is dependent on the molar ratio between ionizable lipids and mRNA nucleotides. Our results show that fractions of ionizable lipids and mRNA (1:1 molar ratio of hEPO mRNA nucleotides:ionizable lipids) of endocytosed LNPs were detected in endo-EVs. Importantly, these EVs can protect the exogenous mRNA during in vivo delivery to produce human protein in mice, detected in plasma and organs. Compared to LNPs, endo-EVs cause lower expression of inflammatory cytokines.

[1] Department of Rheumatology and Inflammation Research, Institute of Medicine, Sahlgrenska Academy, University of Gothenburg, 413 46 Gothenburg, Sweden. [2] Advanced Drug Delivery, Pharmaceutical Sciences, BioPharmaceuticals R&D, AstraZeneca, Gothenburg, 431 83 Mölndal, Sweden. [3] Translational Biomarkers and Bioanalysis, Clinical Pharmacology & Safety Sciences, BioPharmaceuticals R&D, AstraZeneca, Gothenburg, 431 83 Mölndal, Sweden. [4] Department of Chemistry and Molecular Biology, University of Gothenburg, Box 462, 405 30 Gothenburg, Sweden. [5] Animal Sciences and Technologies, Clinical Pharmacology & Safety Sciences, BioPharmaceuticals R&D, AstraZeneca, Gothenburg, 431 83 Mölndal, Sweden. Correspondence and requests for materials should be addressed to H.V. (email: hadi.valadi@gu.se)

RNA-based therapeutics, which function by either silencing pathological genes through delivery of siRNA or expressing therapeutic proteins through the delivery of exogenous mRNA to cells, hold great potential for the treatment of various diseases. RNA therapy provides potential new treatment options in multiple diseases and has been tested in clinical trials of several diseases including cancer, infectious diseases, and various inherited genetic diseases[1,2]. However, RNA therapy also faces substantial challenges. For instance, RNA is highly unstable in extracellular fluids because of the presence of nucleases and the fact that mRNA/siRNA needs to be taken up by the right cells and must be able to escape the endosomes to be translocated into the cytosol, for protein expression or gene silencing to occur. The mRNA modifications can increase stability to some extent; however, the transport of mRNA/siRNA to the cytoplasm of recipient cells requires safe and efficient delivery vehicles[3–5].

Lipid nanoparticles (LNPs), currently represent the most advanced platform for RNA delivery[6–13], which have now advanced into human clinical trials[7,14–16], and their mRNA delivery safety profiles have been evaluated in human and non-human primates[17,18]. LNP-mediated mRNA delivery has been tested in preclinical studies of Fabry disease (X-linked lysosomal storage disease) in non-human primates[19] and disease models of Friedreich's ataxia and methylmalonic acidaemia[20,21], metabolic and behavioral abnormalities in a murine model of citrin deficiency[22], preclinical and clinical trials of immunogenicity for protection against Zika and influenza viruses[23,24]. However, their limited capacity to undergo endosomal escape limits the use of LNPs as RNA delivery vehicles, as only a small fraction of RNA efficiently escapes endosomes to reach the cytoplasm of cells[4]. Despite the fact that, a major proportion of LNPs (95%) is endocytosed (taken up) by cells within half hour[6], it is estimated that <2% of the siRNA administered via LNPs escapes the endosomes to reach the cytosol[6,12]. Thereafter, the fate of endocytosed LNPs for example, how and why endosomal escape of LNP-delivered RNA is only in small amounts to reach the cytoplasm, is not completely understood.

Extracellular vesicles (EVs) are a heterogeneous population of nano- and micro-sized vesicles, including microvesicles, exosomes and several other EV populations classified by ISEV[25]. The best described EVs are the exosomes (40–120 nm), which originate from the endosomes and are secreted through the exocytosis pathway[26,27]. The components of cells could be sorted into the intraluminal vesicles of the late endosomes (also called multivesicular bodies; MVBs). Exosomes are then released into the extracellular environment upon fusion of MVBs with the plasma membrane. EVs can be isolated by several methods, but differential ultracentrifugation remains a gold standard method for processing large volumes of cultured supernatants[28–30].

In 2007, we showed for the first time that EVs contain a substantial amount of RNAs, and that EVs transport RNA between cells as a mechanism of genetic exchange[31]. Since EVs act as endogenous carriers for the transfer of RNA between cells, these vesicles could be tailored as siRNA delivery vehicles[32]. EVs can mediate inter-organ communication and deliver cellular cargo between various organs[33–36]. Therefore, EVs are promising in vivo delivery carriers for siRNA-based therapies[36–39]. However, because of their small size, inserting exogenous mRNA into EVs aiming for expressing new proteins remains challenging.

In the current study, we investigated the intracellular fate of LNP-delivered modified mRNA encoding human erythropoietin (hEPO protein), as well as the ionizable cationic lipid components (DLin-MC3-DMA and DLin-DMA) of LNPs. As stated above, <2% of the siRNA administered via LNPs escapes the endosomes, we hypothesized that remaining part of the LNP materials, i.e. hEPO-mRNA and ionizable lipids, which are localized to

endosomes[6], can be incorporated into intraluminal vesicles of the late endosomes, and could subsequently be secreted into the extracellular environment as EVs. EVs (endosomal and plasma membrane origin) secreted after the endocytosis of LNP-mRNA are referred endo-EVs.

The present study shows that LNP components (mRNA and ionizable lipids) are partly incorporated into endo-EVs, and that the exogenously delivered mRNA is detected at a molar ratio of 1:1 (mRNA nucleotides: ionizable lipids). i.e. the mRNA should be neutrally charged by ionizable cationic lipids to enable mRNA escape, from negatively charged endosomal membrane, to reach the cytoplasm. Most importantly, these endo-EVs protect exogenous mRNA during in vivo transport to organs, and deliver the intact hEPO-mRNA to the cytoplasm of recipient cells. What comes more important is that the delivered mRNA is functional and produces human EPO protein in mice. Although the systemic delivery of both EVs and LNPs cause the expression of proinflammatory cytokines in mice, the expression levels induced by EVs are not as much higher as LNPs did.

We believe that our data could impact the production of alternative biological vehicles for the delivery of mRNAs to express proteins which are absent in the host with genetic disorders, as exemplified herein by the delivery of hEPO mRNA encoding hEPO, a secretory protein important for treating various anemic disorders.

## Results

**Characterization of LNPs**. The LNP formulations were characterized with respect to several biophysical parameters including loading efficiency, average size, polydispersity index (PI) and molar ratio between individual components of LNPs (Supplementary Table 1). The efficiency of hEPO-mRNA constructs in LNPs, defined as efficiency of encapsulation (EE) was 93–97%, and the average size of LNPs with mRNA (LNP-mRNA) varied between 82–90 nm. The concentration of hEPO-mRNA in LNPs was 0.1 mg/mL. The molar percentage ratios between individual components of LNPs using the ionizable lipid DLin-MC3-DMA or DLin-DMA were as follows: ionizable lipid/cholesterol/DSPC/DMPE-PEG: 50/38.5/10/1.5. The chemical structures of the two ionizable lipids are presented in Supplementary Fig. 1a. LNPs containing DLin-MC3-DMA ionizable lipids are defined as MC3-LNPs in this study, whereas LNPs containing DLin-DMA ionizable lipids are defined as DD-LNPs.

**Delivery of hEPO mRNA to cells via LNPs**. The delivery of mRNA encoding human erythropoietin (hEPO protein) to cells was investigated using two different formulations of LNPs i.e. DD-LNPs and MC3-LNPs. The efficacy of mRNA delivery was examined by determining the intracellular amount of hEPO mRNA and the amount of hEPO protein produced.

The hEPO mRNA (100 µg) was transferred to cells via LNPs. After 96 h of LNP administration, hEPO mRNA was quantified in the lysates of recipient cells, and the hEPO protein was quantified both in cell lysates and the supernatants of cell-conditioned media. The results demonstrated that both formulations of LNPs could deliver hEPO mRNA to cells (Fig. 1a) and cause the production of hEPO protein (Fig. 1b, c). Compared to DD-LNPs, MC3-LNPs delivered significantly higher amounts of hEPO mRNA to cells, and accordingly produced higher amounts of hEPO protein. As hEPO is a secretory protein[40,41], it was mostly detected in the extracellular fractions (Fig. 1c) compared with the levels detected in cell lysates (Fig. 1b).

Nanoparticles cause cellular stress[42–44], and activate the autophagic-lysosomal pathway[45], which in current study may depend on the chemical composition of LNPs, as well as the

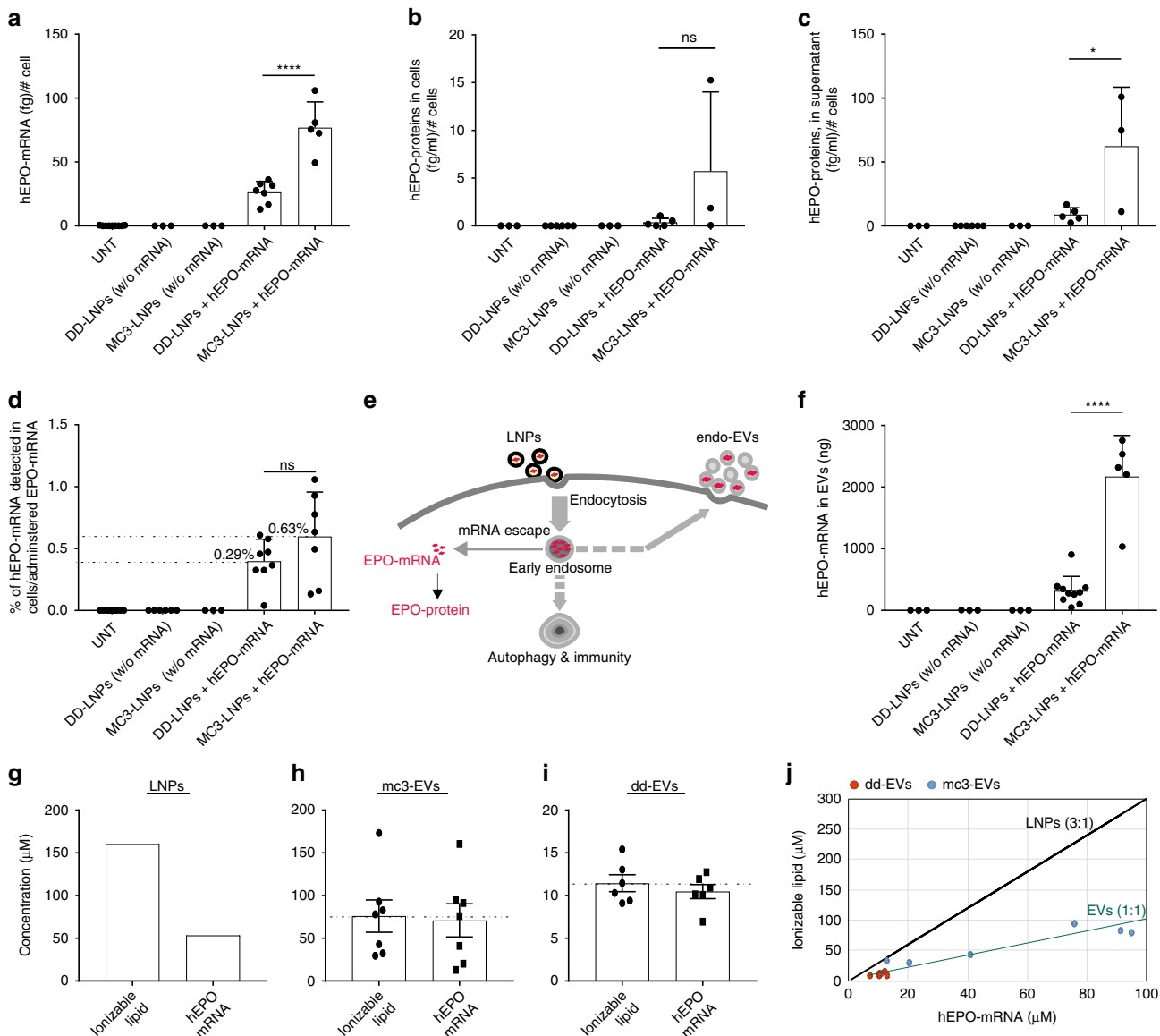

**Fig. 1** Delivery of hEPO mRNA to cells via LNPs and analysis of endo-EVs. MC3-LNPs and DD-LNPs containing 100 μg of hEPO-mRNA were transferred to human epithelial (HTB-177) cells. Untreated cells and cells treated with empty LNPs (without hEPO mRNA) were used as controls. **a** Amount of hEPO mRNA detected in cells. Untreated ($n = 9$), DD-LNPs (w/o mRNA) ($n = 3$), MC3-LNPs (w/o mRNA) ($n = 3$), DD-LNPs + mRNA ($n = 7$), and MC3-LNPs + mRNA ($n = 5$). **b** Amount of hEPO protein detected in cells. Untreated ($n = 3$), DD-LNPs (w/o mRNA) ($n = 6$), MC3-LNPs (w/o mRNA) ($n = 3$), DD-LNPs + mRNA ($n = 5$), and MC3-LNPs + mRNA ($n = 3$). **c** Amount of hEPO protein detected in the supernatant of cultured cells. Untreated ($n = 3$), DD-LNPs (w/o mRNA) ($n = 6$), MC3-LNPs (w/o mRNA) ($n = 3$), DD-LNPs + mRNA ($n = 5$), and MC3-LNPs + mRNA ($n = 3$). **d** Percentage of hEPO mRNA detected in the cytosol of cells relative to the total amount of mRNA administered (100 μg) to cells via LNPs. Untreated ($n = 10$), DD-LNPs (w/o mRNA) ($n = 6$), MC3-LNPs (w/o mRNA) ($n = 3$), DD-LNPs + mRNA ($n = 8$), and MC3-LNPs + mRNA ($n = 7$). **e** Hypothetical presentation of the endosomal escape of hEPO mRNA of LNPs into the cytoplasm and translation into protein, versus loading of hEPO mRNA into endo-EVs. **f** Total amount of hEPO mRNA quantified in endo-EVs isolated from LNP-treated cells. Untreated ($n = 3$), DD-LNPs (w/o mRNA) ($n = 3$), MC3-LNPs (w/o mRNA) ($n = 3$), DD-LNPs + mRNA ($n = 10$), and MC3-LNPs + mRNA ($n = 5$). **g** Molar concentrations of ionizable lipids and hEPO-mRNA (ionizable lipids per hEPO mRNA nucleotides) in originally formulated LNPs (control, $n = 1$), which contains 3 moles of ionizable lipids per 1 mole of mRNA nucleotides. **h** Molar concentration of ionizable lipids and hEPO-mRNA of mc3-EVs ($n = 7$). **i** Molar concentration of ionizable lipids and hEPO-mRNA of dd-EVs ($n = 6$). **j** Stoichiometric comparison between LNPs and endo-EVs regarding molar ratio (mole/mole) of ionizable lipids per hEPO-mRNA nucleotides. Red circles (dd-EV) and blue circles (mc3-EV). Ionizable lipids and hEPO-mRNA of mc3-EVs ($n = 7$) each. Ionizable lipids and hEPO-mRNA of dd-EVs ($n = 6$) each. Data are presented as scatter dot plots including the mean (bars) and standard deviation (SD) of the number ($n$) of biologically independent samples specified for each panel. MC3-LNP and DD-LNP groups (**a–d**, **f**) were compared using the unpaired two-tailed Student's *t*-test. **p < 0.01$, ****p < 0.0001$ and ns = not significant. Source data are provided as a Source Data file

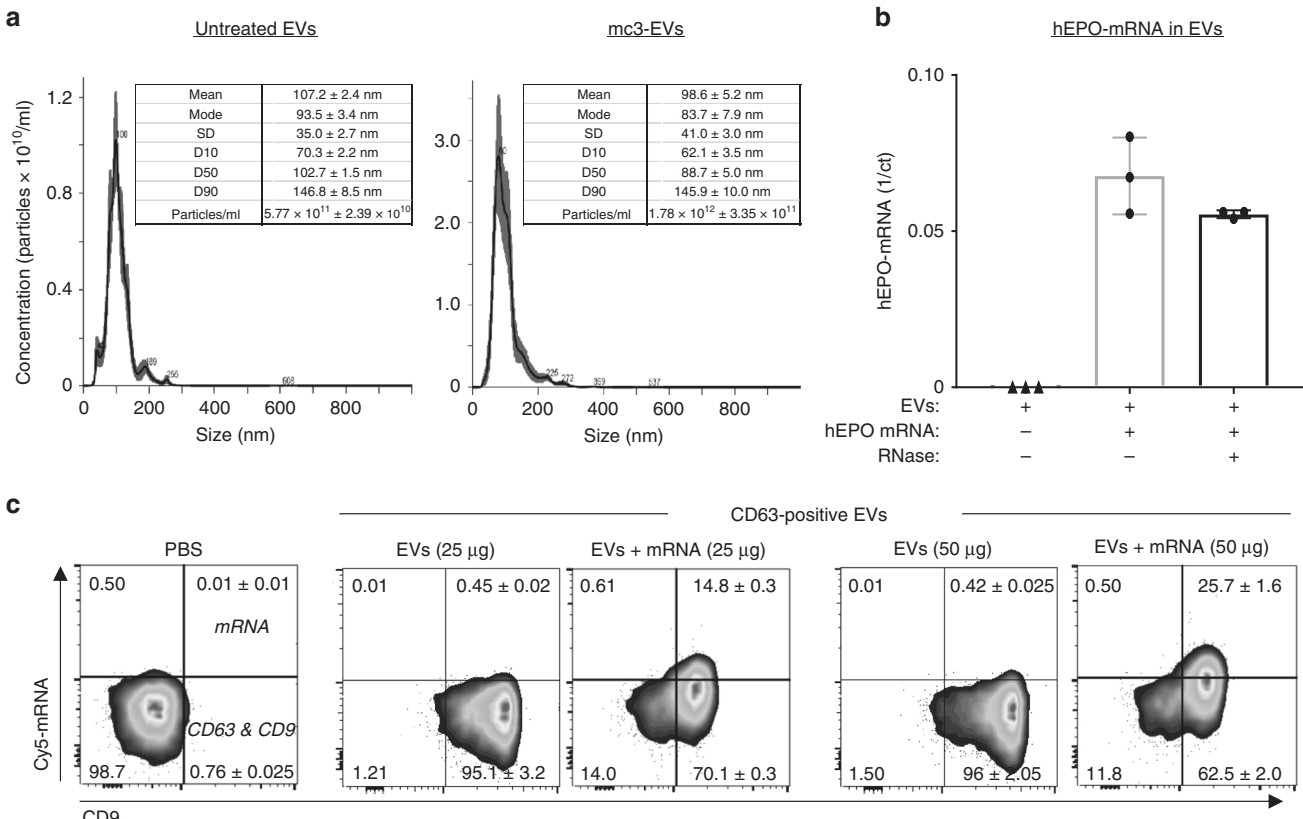

**Fig. 2** Characterization of endo-EVs derived from LNP-treated cells. **a** Nanoparticle tracking analysis for size distribution and concentration of EVs from untreated cells (Left side panel) and from MC3-LNP-treated cells (Right side panel). For each graph (one representative replicate), a table is provided, including mean and mode sizes, SD, and D-values (D10, D50, D90) and particle concentration. **b** EV-mRNA protection assay against RNase A. The hEPO-mRNA qPCR data is represented as scatter dot plot and mean SD of $n = 3$ biologically independent samples. **c** Cy5 mRNA in CD63/CD9 positive EVs. CD63[+] EVs were stained against CD9 antibody and analyzed by FACS for Cy5 mRNA detection. The sole beads incubated with PBS instead of EVs are shown as negative control. Approximately 96% of immunoprecipitated EVs (50 μg assay) from untreated cells are positive for CD63 and CD9 but they are negative for mRNA. By contrast, ~88% of immunoprecipated EVs (50 μg assay) from LNP-treated cells are positive for CD63 and CD9, and 26% EVs contain mRNA that is secreted after the endocytosis of LNP-mRNA. The percentage of CD63/CD9 positive EVs containing Cy5 mRNA is presented in the upper right quadrant. The FACS dot plots represent Cy5 mRNA (y-axis) vs. CD9 (x-axis). One out of two biological replicates is shown. Source data are provided as a Source Data file

modified mRNA. We therefore, investigated the effects of the LNPs on cells. Compared with untreated cells, the treatment with mRNA-loaded LNPs increased the generation time (retarded cell growth), which remained unaffected when cells received empty LNPs (LNPs without hEPO mRNA) (Supplementary Fig. 1b). LNP treatment had no effect on the amount of total RNA per cell (Supplementary Fig. 1c). Total amount of intracellular proteins remained unchanged after LNP (MC3- or DD) treatment (Supplementary Fig. 1d). By contrast, the total amount of extracellular proteins increased after treatment with mRNA-loaded DD-LNPs compared with untreated cells or cells treated with empty DD-LNPs (Supplementary Fig. 1e).

**Characterization of EVs.** Assessment of the effect of hEPO mRNA loaded LNPs on cell-derived EVs showed that the total RNA content of EVs from cells treated with mRNA-loaded MC3-LNPs was higher than EVs from cells treated with mRNA-loaded DD-LNPs or untreated cells (Supplementary Fig. 2a). Total EV protein is slightly increased after MC3-LNP treatment (Supplementary Fig. 2b).

Additionally, EVs were characterized for their size and concentration. The mean ± SEM of the EVs mode size (measured from triplicate samples) was $116.4 ± 9$ nm from MC3-LNP-

treated cells and $100.2 ± 4$ nm from untreated cells (Fig. 2a, Supplementary Fig. 2c, d). The mean ± SEM concentration of EVs was $5.81 × 10^{11} ± 2.25 × 10^{10}$ particles/ml from untreated cells, and $1.25 × 10^{12} ± 1.54 × 10^{11}$ particles/ml from MC3-LNP-treated cells.

Previous studies have used siRNA delivery and showed that only a small fraction of the siRNA (<2%) escapes from endosomes[8,46]. However, in the present study we investigated the fate of LNP-delivered mRNA and other components of LNPs after their uptake by cells, which showed that <1% of the administered mRNA by LNPs was detected in the cytosol of LNP-treated cells (Fig. 1d). Moreover, mRNA delivered via MC3-LNPs undergoes endosomal escape at approximately two-fold higher rate than that delivered by DD-LNPs.

We hypothesized that there could be a link between endocytosis (uptake of LNPs) and exocytosis, wherein, some of the LNP components could be packaged into EVs originated from endosomes (Fig. 1e). EVs (endosomal and plasma membrane origin) secreted after the endocytosis of LNPs are referred endo-EVs. Endo-EVs obtained from MC3-LNP-treated cells are defined mc3-EVs in this study, whereas endo-EVs obtained from DD-LNP-treated cells are defined dd-EVs. The hEPO mRNA in endo-EVs from LNP-treated cells was quantified by qPCR. The results showed that endo-EVs contained hEPO mRNA, and the mRNA

levels were 10-fold higher in endo-EVs secreted from MC3-LNP-treated cells compared to DD-LNP-treated cells (Fig. 1f). Of particular note, cells treated with hEPO mRNA-loaded LNPs contained both hEPO mRNA and hEPO protein (Fig. 1a–c), however, EVs derived from such LNP-treated cells contained hEPO mRNA (Fig. 1f), but not the hEPO protein.

To determine whether the interaction between LNPs and EVs could also have occurred independent of cells and whether mRNA was transferred to EVs outside of cells, the LNPs containing hEPO mRNA were mixed with EVs isolated from the supernatant of untreated cell cultures. After 2 h of incubation, EVs were re-isolated and tested for the presence of hEPO mRNA. The results showed that EVs were negative for hEPO mRNA when directly mixed with LNPs in the absence of cells (Supplementary Fig. 3), indicating that LNPs do not transfer hEPO mRNA directly to EVs outside of cells, but processed inside the cell. This is further supported by the difference between mRNA nucleotides and ionizable lipids ratio in originally formulated LNPs (1:3, see methods) and detected in endo-EVs (1:1) (Fig. 1g–j), which would be expected to be the same (3:1) if direct fusion of LNPs and EVs had occurred.

Additionally, to show that mRNA is located/protected inside EVs, the mc3-EVs containing hEPO-mRNA were exposed to RNase treatment, total RNA was isolated from EVs and hEPO-mRNA was quantified by qPCR. The results showed that, despite the efficient endonucleolytic activity of the RNase (shown on EV free RNA), only a 2 Ct fold-change decrease in the hEPO-mRNA content was observed in the RNase treated mc3-EVs than untreated mc3-EVs (Fig. 2b, Supplementary 2e). Although it is not possible to exclude that a small portion of the delivered mRNA could be attached to the external membrane layer of the EVs, most hEPO-mRNA is protected from the endonucleolytic activity.

To further show that LNP-mRNA is carried by EVs, the mc3-EVs pellets isolated from MC3-LNP-treated cells were immuno-precipitated with CD63 antibody, and examined for the presence of Cy5 mRNA in EVs. The FACS analysis showed that approximately 96% of immunoprecipitated EVs (50 μg assay) from untreated cells were positive for CD63 and CD9 but they were negative for the mRNA. By contrast, approximately 88% of immunoprecipated EVs (50 μg assay) from LNP-mRNA-treated cells were positive for CD63 and CD9 and 26% contain mRNA (Cy5 mRNA) that is secreted after the endocytosis of LNPs containing Cy5 mRNA (Fig. 2c). Notably, in the negative controls (only beads and CD63$^+$/CD9$^+$ EVs from untreated cells) no Cy5 mRNA signal was detectable, confirming that mRNA is carried by CD63$^+$/CD9$^+$ EVs of LNP-treated cells.

After detecting hEPO mRNA in EVs, we further analyzed whether the LNP components i.e. cationic ionizable lipids are also present in endo-EVs after treating cells with LNPs. We discovered that EVs secreted from LNP-treated cells also contain ionizable lipids. The originally formulated LNPs, which were transferred to cells contained 3 moles of ionizable lipid per one mole of mRNA nucleotides (see methods); a 1:3 (mol/mol) ratio of hEPO mRNA nucleotides/ ionizable lipids for both MC3- and DD-LNPs (Fig. 1g). However, endo-EVs contained stoichiometrically less ionizable cationic lipids per hEPO mRNA nucleotides that is 1:1 (1 mol of ionizable lipid per one mole of mRNA nucleotides) (Fig. 1h, i). The molar ratios between ionizable lipids and mRNA in both LNPs and in EVs are plotted (Fig. 1j). This suggests that the molar ratio between ionizable cationic lipid and mRNA nucleotides should be 1:1 (neutrally charged RNA-lipid complex) to enable mRNA escape, from negatively charged endosomal membrane, to reach the cytoplasm.

In UPLC-MS analysis, the DLin-MC3-DMA samples were dissolved in either ethanol, 1% (w/w) Triton X-100 or in 1%

(w/w) Triton-X100 spiked with a fixed amount of representative sample of untreated-EVs and injected on the UPLC-MS system. The response for the Triton X-100 (with and without untreated-EVs) samples is lower compared to the samples dissolved in pure ethanol, but we estimate the error in quantification to be less than 10% (Supplementary Fig. 4). Considering the likely variation in the sample preparations (for mRNA and lipid quantifications) and the actual qPCR analysis (mRNA) and UPLC-MS (lipids) the observed error does not influence the main conclusion that mRNA and the ionizable lipid are co-transported as a complex stochiometric salt (1:1) into the secreted EVs. Furthermore, we observe a very strong correlation between nucleotide concentration measured by qPCR and lipid concentration determined by UPLC-MS for the different EV samples (Fig. 1j).

**Delivery of mRNA to epithelial and immune cells via endo-EVs.** After having detected mRNA in endo-EVs, we investigated whether endo-EVs could transport the exogenous mRNA to recipient cells, acting as RNA delivery vehicles.

For tracking the mRNA uptake by recipient cells, a labelled mRNA (Cy5 mRNA) was delivered via endo-EVs to four different cell types; HTB-177, B-cells, T-cells and monocytes. Immune cells are generally difficult to transfect with RNA using other delivery vehicles. Peripheral blood mononuclear cells (PBMCs) isolated from buffy coats of healthy humans and HTB-177 cells were separately incubated with dd-EVs containing Cy5 mRNA. After incubation, PBMCs were harvested and stained with monoclonal antibodies (mAbs) against surface markers for B-cells, T-cells and monocytes. Flow cytometric analysis detected Cy5 mRNA in recipient cells as early as 5 h after EV-mediated mRNA delivery (Supplementary Figs. 5a–d, 6).

At 24 h of EV delivery, 70% of the HTB-177 cells were positive for Cy5 mRNA, whereas at 48 h, 40% of cells were Cy5 mRNA positive (Supplementary Fig. 5a). Compared to HTB-177 cells, the uptake in B-cells was lower, with 6% of B-cells positive for Cy5 mRNA at 5 h, and 30% and maximum 40% positive cells after 24 and 48 h, respectively (Supplementary Fig. 5b). T-cells displayed lower uptake, with maximum 14–17% of T-cells positive for Cy5 mRNA during the evaluation period (Supplementary Fig. 5c). By contrast, monocytes exhibited the highest uptake, as 71% of monocytes were positive for Cy5 mRNA at 5 h, with a decrease to 60 and 40% positive cells at 24 and 48 h, respectively (Supplementary Fig. 5d).

After having confirmed that EVs can deliver mRNA to different cell types, we investigated whether endo-EVs could transport the functional mRNA i.e. exogenous hEPO mRNA to express exogenous hEPO protein in recipient cells. The mc3-EVs and dd-EVs containing hEPO mRNA were transferred to HTB-177 cells. Analysis revealed that hEPO protein is expressed in cell lysates and is secreted in the supernatants of recipient cells in vitro (Supplementary Fig. 5e–g). The hEPO mRNA delivered by mc3-EVs produced higher amounts of hEPO protein than dd-EVs (Supplementary Fig. 5g).

We examined whether EV-based delivery acts differently than LNPs on cellular behavior. The results showed that EV-mediated delivery does not affect the cell generation time or cellular protein amounts, regardless if EVs carry hEPO mRNA or are without hEPO mRNA (Supplementary Fig. 5h–j).

**Delivery of human EPO mRNA to mice via endo-EVs.** Since endo-EVs could deliver hEPO mRNA to cells in vitro and cause the production of hEPO protein in recipient cells, we investigated whether these EVs could deliver hEPO-mRNA to cells in vivo and produce human protein in mice. First, the cross-reactivity of mouse EPO protein against human EPO antibodies was examined

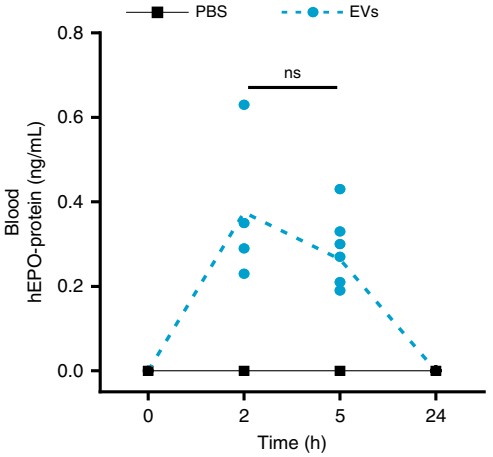

**Fig. 3** Detection of hEPO protein in mouse blood after hEPO mRNA delivery via EVs Mice were intravenously injected with 100 μL of mc3-EVs containing 1.5 μg of hEPO mRNA (per mouse). The concentrations of hEPO protein in murine plasma were determined by Gyros immunoassay for hEPO at 0 (untreated), 2, 5, and 24 h after EV injection. The hEPO protein was detected in mouse blood at 2 h after EV injection. $N = 8$ independent animals at each time point except for 2 h ($n = 4$) are presented. The plasma hEPO protein from mc3-EV delivery was compared between 2 and 5 h by unpaired two-tailed Student's t-test, although the difference was not statistically significant (ns = not significant). Source data are provided as a Source Data file

using a human erythropoietin ELISA assay in plasma from untreated mice, which did not produce a signal for mouse EPO (Supplementary Fig. 7). Therefore, in our experiments if we detect a background noise from mouse EPO, it would not interfere with our results. For all in vivo experiments, control mice injected with PBS were included to assess for possible background levels of mouse EPO.

C57BL6/NCrl mice were injected intravenously with a single dose of mc3-EVs (1.5 μg of hEPO mRNA per mouse), and the production of hEPO protein was examined in plasma and organs. The hEPO protein was detected in mouse plasma after 2 h of EV-mediated hEPO mRNA delivery, indicating that EVs could deliver exogenous mRNA and can produce the protein (Fig. 3).

The presence of hEPO mRNA and hEPO protein was examined in eight organs of mice sacrificed at 5, 24 and 96 h after EV injection. The results showed that EVs not only delivered hEPO mRNA to different organs, but also caused the production of hEPO protein (Fig. 4). The hEPO protein was detectable in four of the organs evaluated, namely, the heart, lung, liver and spleen (Fig. 4a–h). In the heart and lung, hEPO mRNA as well as hEPO protein were detectable after 5 h of EV-mediated mRNA delivery (Fig. 4a–d). In the liver, hEPO mRNA was only detectable after 5 h, whereas hEPO protein was detectable at 5 and 24 h (Fig. 4e, f). A different pattern was observed in the spleen, where hEPO protein was only detectable at 5 h (Fig. 4g, h).

Notably, among the four organs positive for hEPO mRNA, the liver had the highest amount of hEPO protein (Fig. 4f), whereas the spleen had the highest amount of hEPO mRNA (Fig. 4g). However, the amount of hEPO protein detected in the spleen at 5 h was comparable to that in other organs. The kidney was positive for hEPO mRNA, whereas the protein levels were in the same range as those in the PBS controls, indicating the presence of background noise and the lack of considerable protein levels in this organ (Fig. 4i, j). In three of the eight organs analyzed (thymus, pancreas, and brain), hEPO protein levels were comparable to those detected in the negative control (PBS) (Fig. 4k–p).

A parallel experiment was performed with MC3-LNPs to examine hEPO mRNA delivery by LNPs. C57BL6/NCrl mice were injected intravenously with a single dose of MC3-LNPs (1.5 μg of hEPO mRNA per mouse), and the production of hEPO protein was examined in plasma. The hEPO protein was detectable in plasma after 2 h of LNP-mediated hEPO mRNA delivery (Supplementary Fig. 8). The hEPO mRNA and hEPO protein were detectable in five organs, heart, lung, liver, spleen and kidney (Supplementary Fig. 9). Among hEPO mRNA-positive organs, the spleen showed the highest amount of hEPO mRNA, which persisted for 96 h, followed by the kidney, in which hEPO mRNA persisted for 24 h. By contrast, most of the hEPO protein was detected in the liver. In the heart and lung, both hEPO mRNA and protein were detected at 5 h after injection. In the thymus, pancreas and brain, the levels of hEPO mRNA and hEPO protein were comparable to those in the corresponding negative samples (PBS).

**Comparison between EVs and LNPs for the delivery of hEPO mRNA to mice.** The MC3-LNP formulations based on DLin-MC3-DMA used in the present study are the most potent lipids currently used in clinical trials. Therefore, we sought to compare how efficiently EVs deliver hEPO mRNA and their biodistribution, compared to MC3-LNPs using the same dose of mRNA. In the lung and liver, the amount of hEPO protein produced was comparable between the two delivery vehicles. However, LNP-based hEPO mRNA delivery resulted in a higher protein production in the spleen and, to some extent, in the heart, than EV-based mRNA delivery. The plasma concentration of hEPO protein was higher in LNP-delivered than in EV-delivered mRNA-treated mice (Fig. 5).

Additionally, the assessment of organ weight showed that MC3-LNPs and mc3-EVs had no effect on tissue weight (Supplementary Fig. 10).

**Expression of inflammatory cytokines against systemic delivery of EVs and LNPs.** DLin-MC3-DMA ionizable lipids are important components in the formulation of most advanced LNPs; however, they are partially immunogenic to recipient cells and elicit immune responses in the host. Since EVs had a lower amount of ionizable lipids than LNPs (1/3), we investigated whether EVs were accordingly less immunogenic to recipient mice. The secreted levels of eight inflammatory cytokines were measured in the plasma of mice at 5 and 24 h of LNPs and EVs injection. Our results indicate that although the systemic delivery of both EVs and LNPs cause the expression of proinflammatory cytokines including IL-6, IP-10, RANTES, MCP-1, and KC in mice, the expression levels induced by EVs were not as much higher as LNPs did (Fig. 6). This indicates that EVs might be better tolerated by the recipient mice, since EVs contain fewer ionizable lipid molecules per mRNA nucleotides (1: 1 molar ratio) compared to LNPs (1: 3 molar ratio).

**Discussion**

LNPs represent one of the most potent RNA delivery vehicles in vivo, and are currently being tested in human clinical trials. However, the fate of the RNA and the other components of LNPs inside cells remains unclear.

In the present study, we showed that the part of the LNP-hEPO mRNA and ionizable lipid components of LNPs, which had not been dissociated/escaped into cytoplasm and which had not been degraded in early endosomes were packed into endo-EVs and secreted outside the cell. The experiment was performed 13 times (seven times with MC3-LNPs and six times with DD-LNPs), and the results consistently showed that EVs contained one mole of

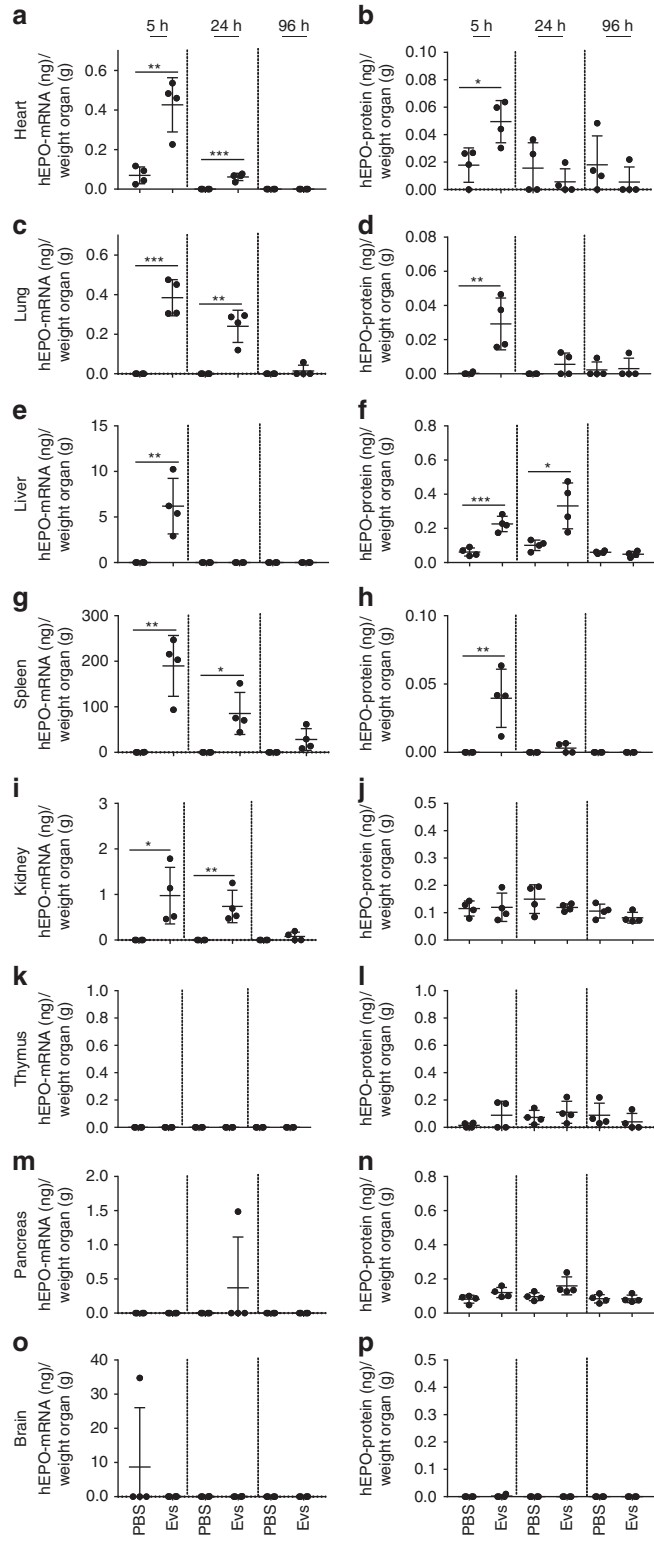

**Fig. 4** Quantification of hEPO mRNA and hEPO protein in mouse organs. Mice were intravenously injected with 100 µL of mc3-EVs containing 1.5 µg of hEPO-mRNA (per mouse). At 5, 24, and 96 h after EV injection, the levels of hEPO mRNA and hEPO protein were determined in eight organs by qPCR and ELISA, respectively. **a**, **b** Levels of hEPO mRNA and protein in the heart, **c**, **d** lung, **e**, **f** liver, **g**, **h** spleen, **i**, **j** kidney, **k**, **l** thymus, **m**, **n** pancreas, and **o**, **p** brain are shown. The highest amount of hEPO protein was detected in the liver, whereas the highest amount of hEPO mRNA was detected in the spleen. Data are presented as the mean (bars) and standard deviation (SD) $n = 4$ independent animals at each time point. EVs and untreated groups were compared at each time point using the unpaired two-tailed Student's $t$-test. *$p < 0.05$, **$p < 0.01$, and ***$p < 0.001$. Source data are provided as a Source Data file

We show that LNPs can transfer exogenous hEPO mRNA to cells and produce new protein, i.e. hEPO in vitro both in the cytosol and secreted in the culture supernatant, and in vivo in organs and in the blood in secreted form (hEPO protein is a secretory protein). MC3-LNPs are more effective for delivering functional hEPO mRNA to cells and produce approximately 8-fold higher amounts of hEPO protein than DD-LNPs for the same dose of hEPO mRNA delivered to cells.

Moreover, the delivery of Cy5-labelled mRNA to the three primary immune cells, including B-cells, T-cells, and monocytes via endo-EVs indicates a high potential for transferring genetic materials (mRNAs) to human blood cells, such as B-cells, which are difficult to transfect.

We compared the efficacy of LNPs and endo-EVs for the delivery of mRNA to eight different organs and peripheral blood. After LNP or endo-EV delivery in vivo, the translation kinetics of hEPO mRNA in plasma and organs at different time points revealed that the hEPO mRNA transferred to mice via endo-EVs is functional. Because mice lack this form of EPO (hEPO), the EPO ELISA kit used could effectively distinguish between mouse and human EPO. The protein is produced from exogenous mRNA, which the recipient mice lack. Compared to endo-EVs, LNPs led to a higher production of hEPO in the organs analyzed in this study, and the most noticeable difference was observed in the plasma and spleen. Quantification of plasma hEPO protein showed that LNPs resulted in a 6–8-fold higher hEPO production than endo-EVs.

The differences in hEPO expression between endo-EVs and LNPs may be due to specific factors, e.g. that not all EVs contain hEPO mRNA. Some EVs taken up by recipient cells may lack hEPO mRNA, as not all EVs secreted from cells in standard preparations carry RNA (one copy of mRNA per 10,000 EVs)[47] and are, therefore, individually some EVs are unlikely to be functional as vehicles for RNA-based delivery[47,48].

Since endo-EVs contained a three-fold lower level of ionizable lipids (which, despite being important compounds for LNP manufacturing, can be toxic) per mRNA than LNPs, we expected that endo-EVs should elicit a milder immune response than LNPs. The expression levels of eight different cytokines in the plasma of mice confirmed our expectation that endo-EVs induce less inflammatory cytokine responses than LNPs upon transferring an equal dose of hEPO mRNA to mice.

Higher levels of pro-inflammatory cytokine responses elicited by LNPs compared to endo-EVs could be related to several factors, e.g. (i) endo-EVs delivered 1/3 of the toxic synthetic ionizable lipids of LNPs, (ii) in contrast to LNPs, EVs are natural biological products, and might be better tolerated by the host, and (iii) the routes of cellular uptake differ between EVs and LNPs, which could behave differently to autophagic-lysosomal pathway. The advantage of using EVs for mRNA delivery would be that

ionizable lipids per mole of hEPO mRNA nucleotides. This differed from LNPs, which contained three moles of ionizable lipids per mole of hEPO mRNA nucleotides. Despite variation in the levels of EPO-mRNA and ionizable lipids in endo-EVs, the molar ratio between them remained the same (1:1). These results together with the results of the cell free mixture of EVs and LNPs indicated that LNPs do not fuse with EVs outside the cell, but rather processed in the endosomal pathway and are secreted in endo-EVs.

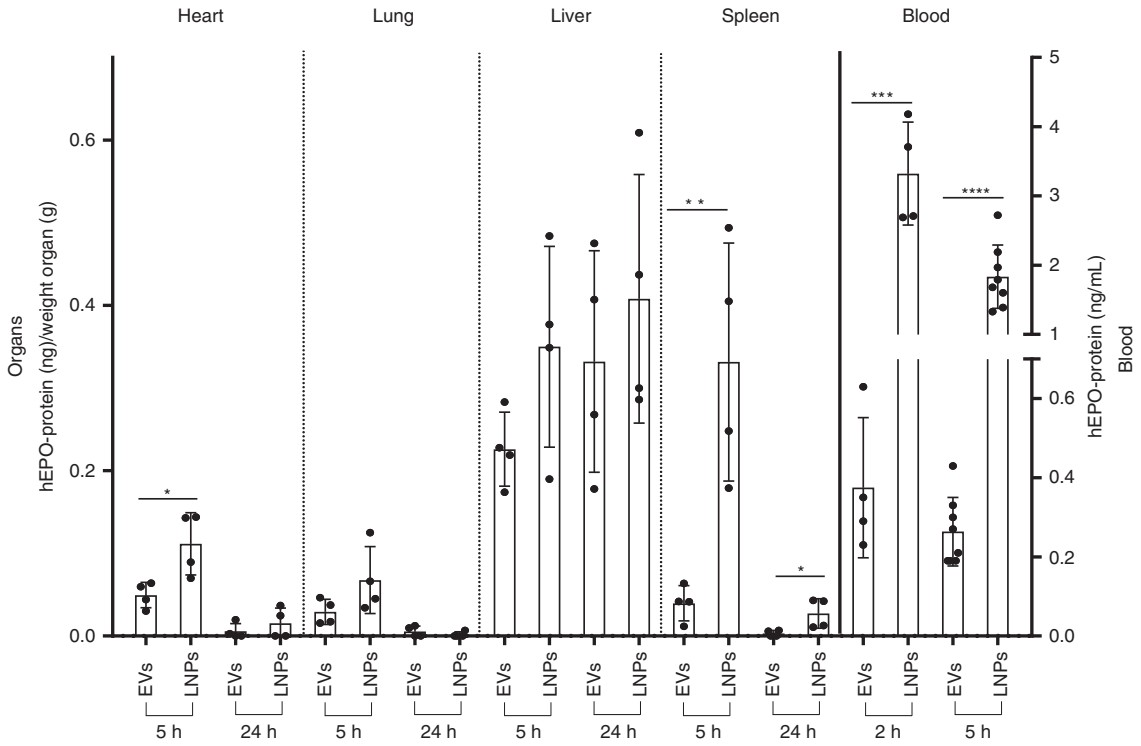

**Fig. 5** Comparison of hEPO protein levels in murine blood and organs. The ability of mc3-EVs and MC3-LNPs to produce hEPO protein upon injection of equal doses of hEPO mRNA (1.5 µg) into mice was compared. In most organs, the amount of hEPO protein was comparable between LNPs and EVs except for the spleen, which showed a significant difference in protein production followed by the heart (less significant difference). The most significant difference was observed in the plasma levels of hEPO protein, which were considerably higher for MC3-LNPs than for mc3-EVs. Data are presented as the mean (bars) and standard deviation (SD) $n = 4$ independent animals at each time point except for LNPs and EVs at 5 h for plasma analysis ($n = 8$). EV and LNP groups were compared for each organ or plasma at each time point using the unpaired two-tailed Student's $t$-test. *$p < 0.05$, **$p < 0.01$, ***$p < 0.001$, and ****$p < 0.0001$. Source data are provided as a Source Data file

compared to synthetic products, EVs are biological products and might elicit a milder immune response in the host.

In the case of LNPs, cellular uptake is mediated by endocytosis, which could activate the cells' autophagic-lysosomal pathway. Accumulating evidence indicates that endocytosis of nanoparticles generates autophagosomes, and their subsequent fusion with lysosomes leads to the digestion of their content[45]. Autophagosomal–lysosomal activation shapes the cellular immunity as a defence mechanism against foreign particles, by which innate immune effectors elicit inflammatory responses[49–51]. In the present study, the uptake of LNPs caused an increased cytokine release. The routes of EV uptake[52–54] differ from those of LNPs and are not likely to elicit the autophagy–lysosomal pathway, as they release their content into the cytoplasm probably without undergoing lysosomal trapping. Additionally, because of their small size, EVs can escape from rapid phagocytosis, and steadily carry and deliver RNA in circulation, passing through the vascular endothelium to the target cells[55].

Once endocytosed, LNP-encapsulated mRNA must escape from endosomes to reach the cytoplasm of recipient cells for translation into protein. It was suggested that late endosome/lysosome formation is essential for the functional delivery of exogenous mRNA[11]. We examined the linkage between endocytosed LNP-mRNA and secretion via EVs, because the intraluminal vesicles of late endosomes (MVBs) are secreted into the extracellular milieu as EVs[26,27,56].

We introduce a hypothetical mechanism explaining the fate of LNP endosomes and how part of the LNP-mRNA could be translocated to the cytoplasm or loaded (intracellularly) into the luminal vesicles of multivesicular endosomes and secreted from the cell via endo-EVs (Fig. 7a). This is consistent with two independent experimental observations. First, we showed that the molar ratio of ionizable lipid to mRNA nucleotide in EVs was 1:1, whereas that inside LNPs was 3:1. Second, the mRNA loading efficiency was correlated with intracellular protein expression, i.e. the loading efficiency was significantly higher using MC3-LNPs than DD-LNPs. This suggests that LNPs added to cells are endocytosed[4,6,13,46] (Fig. 7a, step 1). Because lysosomes fuse with early endosomes, there is an acidification of the endosomal environment (pH 5.5–6.2)[6] (Fig. 7a, step 2). As the endosomal pH decreases, the LNP surface becomes positively charged and interacts with the negatively charged endosomal membrane[9,57] (Fig. 7a, step 3). Assuming that LNP surface components (e.g. ionizable lipids) are miscible with membrane lipids, LNPs will fuse with the endosomal membrane. Inside LNPs, the ionizable lipids at the endosomal pH are likely to form a water-insoluble complex salt with the mRNA (1:1)[58]; therefore, the lipid-mRNA complex is net neutral and firmly held together. Such a complex salt (1:1 mRNA: ionizable lipids) can be lipid-soluble, and could therefore be transported along the endosomal membrane during the LNP fusion event. Once on the cytoplasmic side of the endosomal membrane, where the pH is neutral (~7.4), the complex will start to dissociate (Fig. 7a, step 4a and Fig. 7b), giving rise to intracellularly available mRNA for protein synthesis. Whereas, at acidic pH the slow dissociation kinetics could be a result of the size of mRNA (Fig. 7b). Therefore, a fraction of the non-dissociated complex (lipid–mRNA) salt could be involved in invagination of the endosomal membrane to form intraluminal

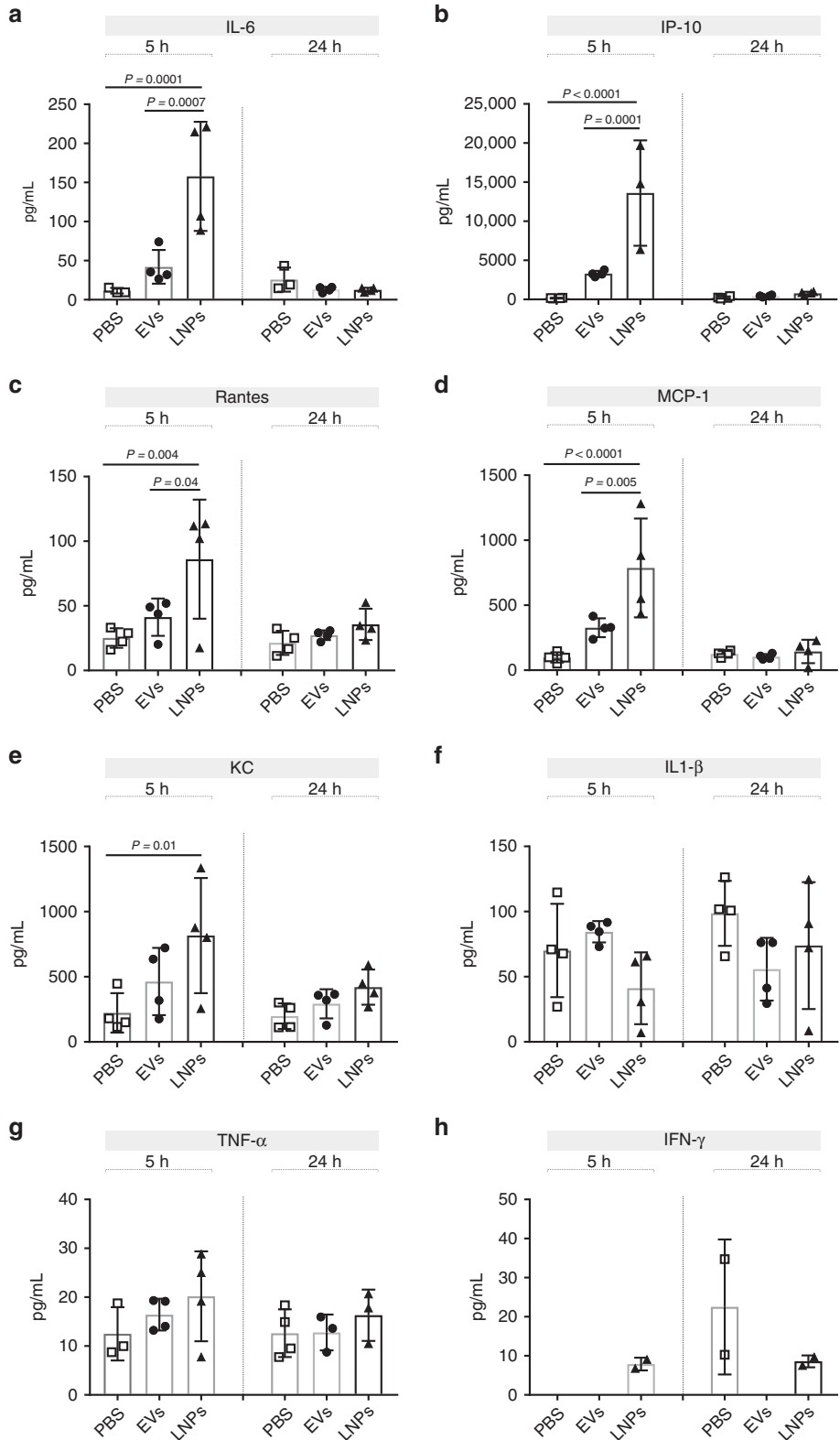

**Fig. 6** Cytokine analysis in mouse plasma after mc3-EV and MC3-LNP delivery. Mice were intravenously injected with 100 µL of mc3-EVs or MC3-LNPs containing 1.5 µg of hEPO-mRNA (per mouse). The concentrations of eight pro-inflammatory cytokines including IL-6 (**a**), IP-10 (**b**), RANTES (**c**), MCP-1 (**d**), KC (**e**), IL1-β (**f**), TNF-α (**g**), and IFN-γ (**h**) were determined in mouse plasma after 5 and 24 h of mc3-EV, MC3-LNP or PBS injection. The levels of pro-inflammatory cytokines were significantly higher in mice receiving LNP injection than in those receiving EV injection. White squares: PBS, black circles: EVs and black triangles: LNPs. Data are presented as the mean (bars) and standard deviation (SD) $n = 4$ independent animals at each time point. Statistical analysis was performed using one-way ANOVA, followed by Sidak's multiple comparisons test. Significant differences are shown as p-values. Source data are provided as a Source Data file

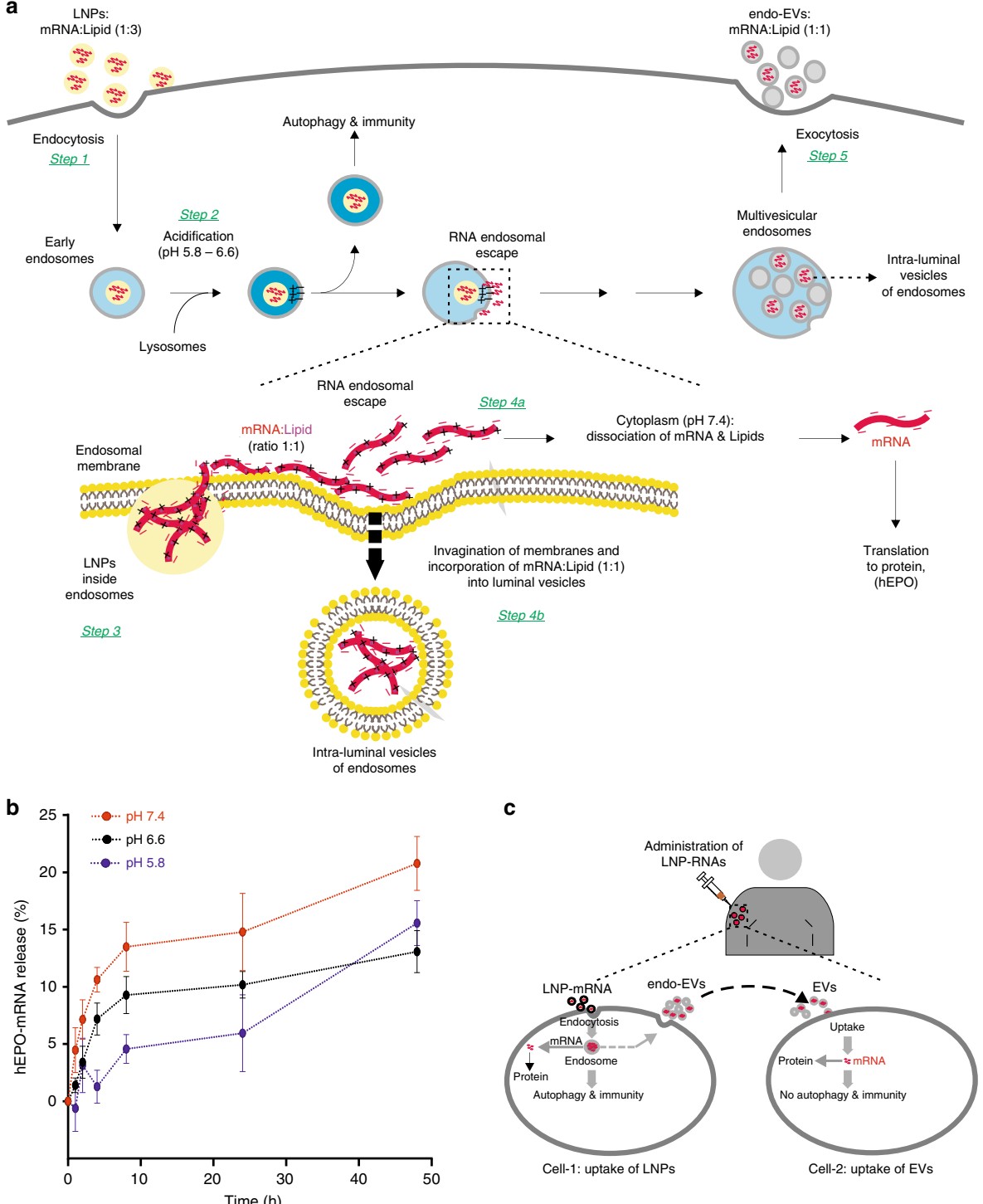

vesicles (Fig. 7a, step 4b) packaged with mRNA. These vesicles are then released into the extracellular environment upon the fusion of MVBs with the plasma membrane and are released outside the cell as EVs (Fig. 7a, step 5). We argue that the stoichiometric ratio between ionizable lipid and mRNA nucleotides should be neutrally charged (1:1) to enable mRNA escape from the endosome engulfing and reach the cytoplasm.

Initially, we and other researchers used external loading methods such as EV electroporation to directly incorporate siRNA into EVs (in the absence of cells), which were used as RNA delivery vehicles[32,37,39,59,60]. Nevertheless, the incorporation of large-sized mRNAs into EVs using internal methods (loading via

cells) was not achieved previously. When DNA plasmids (vectors) are transfected into cells via viruses or liposomes, their transcript RNAs can be detected in EVs secreted by transfected cells. In this model, the DNA plasmid transfected into cells must first be translocated to the nucleus and transcribed into RNA, which then enters the cytoplasm via a nuclear pore complex (functional RNA). A small fraction of this vector RNA expressed in the nucleus can be detected in EVs[61–63].

Here we present an alternative model to load exogenous mRNA into EVs, where no extra route (i.e. nuclear route) is needed. Our method exploits the direct link for transport of molecules between endocytosis of LNPs containing mRNA and

**Fig. 7** A hypothetical mechanism explaining the fate of LNP endosomes. **a** step 1, 2: after the endocytosis of LNPs, lysosomes fuse with early endosomes and cause the acidification of the endosomal environment (pH 5.5–6.2). **a** step 3: the surface of LNPs is positively charged, drawing LNPs to the inner membrane of the endosomes, which is negatively charged[9, 57]. This enables the lipid components of LNPs to fuse with the endosomal membrane, allowing the mRNA translocation to the water phase outside the endosomes. Only the mRNA when neutrally charged by ionizable cationic lipids (ratio 1:1 mRNA: lipid) can cross the endosomal membrane. RNA:lipid ratio other than 1:1 would theoretically be unable to cross the endosomal membrane. **b** In acidic environment (pH 5.8 or 6.6), the mRNA is slightly released from LNPs, whereas at neutral pH (~7.4), the mRNA and lipids are dissociated (The data are shown as standard error of the mean of three replicates). **a** step 4a: part of the LNP-mRNA that escapes the endosomal membrane and localizes to the cytoplasm could be dissociated from the ionizable lipids because the pH of the cytoplasm is neutral, consistent with the results shown in **b**). By contrast **a** step 4b: when LNP-mRNA is transported to the cytoplasmic side of the endosomal membrane, intraluminal vesicles are formed by invagination of the endosomal membrane, and a portion of the LNP-mRNA could be incorporated into these vesicles. **a** step 4a and 5: since only a 1:1 ratio (neutral) can cross the endosomal membrane and become incorporated into luminal vesicles of endosomes, endo-EVs contained a 1:1 ratio of hEPO mRNA and ionizable lipids. **a** step 5: the luminal vesicles are then released into the extracellular environment upon the fusion of multivesicular endosomes with the plasma membrane. **c** Since LNPs with the same ionizable lipids used in this study are currently being utilized in clinical trials and endo-EVs contained hEPO mRNA acquired after the endocytosis of LNPs and delivered to other cells, we postulate that a similar scenario may occur in individuals administered with LNPs, suggesting that part of the mRNA delivery is achieved by such EVs

exocytosis, where endo-EVs acquire different LNP molecules, such as mRNA and ionizable lipids, directly from the endosomal pathway (Figs. 1e, 7a).

LNPs with the same ionizable lipid used in the present study are currently being tested in human clinical trials. As shown in the present study, a similar scenario may occur in humans: when mRNA is delivered via LNPs, the LNPs alone may not deliver mRNA to all cells that express the protein; part of the RNA delivery may be achieved via endo-EVs secreted by cells that internalize the LNPs (Fig. 7c). Additional studies are needed to determine how much of the LNP-delivery is actually achieved by the LNPs' own contribution and not from the endo-EVs originating from LNP-treated individuals.

## Materials and methods

**Formulation and characterization of LNPs.** DLin-MC3-DMA and DLin-DMA LNPs containing modified hEPO-mRNA (858 nucleotides) (5meC, Ψ) (Trilink) were prepared by precipitating the mRNA with four different lipid components as described previously[64]. These components consist of an ionizable lipid; DLin-MC3-DMA or DLin-DMA, which are ionizable (cationic) at low pH, two helper lipids (DSPC and Cholesterol) and a PEGylated lipid (DMPE-PEG2000). A solution of hEPO-mRNA in water was prepared by mixing mRNA dissolved in MilliQ-water, 100 mM citrate buffer pH = 3 and MilliQ-water to give a solution of 50 mM citrate. Lipid solutions in ethanol (99.5%) were prepared with a composition of four lipid components [Ioniziable Lipid:DSPC:Cholesterol:DMPE-PEG2000] = 50:38.5:10:1.5 mol% and a total lipid content of 12.5 mM. The mRNA and lipid solutions were mixed in a NanoAssemblr (Precision Nanosystems) microfluidic mixing system at a volume mixing ratio of Aq:EtOH = 3:1 and a constant total flow rate of 12 mL/min. At the time of mixing, the ratio between the nitrogen atoms on the ionizable lipid and phosphor atoms on the mRNA chain was 3:1. If "empty" LNPs were prepared, i.e. LNPs without any mRNA, the ethanol phase was mixed with only 50 mM citrate buffer pH = 3. The initial 0.35 mL and the last 0.05 mL of the LNP solution prepared were discarded while the rest of the volume was collected as the sample fraction.

In some preparations of LNPs, Cy5-EGFP-mRNA (996 nucleotides) (5meC, Ψ) (Trilink) was loaded instead of hEPO mRNA for separate experiments.

For the characterization of formulated LNPs, following preparation, 25 µL of the sample fraction was injected into 975 of 10 mM phosphate buffer (pH 7.4) and used to measure the intensity-averaged particle size (Z-average) on ZetaSizer (Malvern Instruments Inc.). The sample fraction was transferred immediately to a Slide-a-lyzer G2 dialysis cassette (10000 MWCO, Thermo Fisher Scientific Inc.) and dialyzed over night at 4 °C against PBS (pH7.4). The volume of the PBS buffer was 650–800x the sample fraction volume. The sample fraction was collected and from this volume 25 µL was injected into 975 µL 10 mM phosphate buffer (pH 7.4) and the particle size was measured once again (post dialysis particle size). The final mRNA concentration and encapsulation efficiency (EE) was measured by Quant-it Ribogreen Assay Kit (Thermo Fischer Scientific).

**Cell culture.** The human epithelial HTB-177 (NCI-H460) cell line purchased from ATCC was cultured according to ATCC guidelines. The RPMI-1640 growth medium (Sigma Aldrich) containing sodium bicarbonate, without sodium pyruvate and HEPES, was supplemented with 10% exosome-depleted fetal bovine serum (FBS) (Sigma), 1% of L-glutamine (Thermo Fisher Scientific) and 1% penicillin-streptomycin (Thermo Fisher Scientific), at 37 °C in the presence of 5% $CO_2$. The heat inactivated FBS was exosome-depleted by ultracentrifugation at 120,000×g for

2 h at 4 °C on an Optima L-100 XP ultracentrifuge with 70Ti rotor (Beckman Coulter) and exosome-depleted supernatant was filtered through 0.2µm filters. The fresh buffy coats from healthy donors were obtained from Sahlgrenska University hospital (Gothenburg, Sweden) and the peripheral blood mononuclear cells (PBMCs) were isolated by density-gradient centrifugation. PBMCs were cultured in complete RPMI-1640 growth medium supplemented with L-glutamine, non-essential amino acids, sodium pyruvate, 1% penicillin-streptomycin, β-mercaptoethanol, 10% exosome-depleted FBS and stimulated with goat Anti-Human IgA/IgG/IgM F(ab′)2 fragments 2.5 µg/mL (Jackson ImmunoResearch Laboratories) and phorbol myristate acetate (PMA)1 µg/mL (InvivoGen).

**hEPO mRNA delivery to epithelial cells via LNPs.** The HTB-177 cells were seeded at a density of $3 \times 10^6$ cells/175 cm$^2$ flask in 30 mL of growth medium. After incubation (adaptation) for 24 h, the cells were treated with 1 mL of DD- or MC3-LNPs containing 100 µg of hEPO mRNA/flask in the presence of 1% human serum (Sigma Aldrich), which was administered in three different doses; Day (1) 200 µL LNPs (20 µg mRNA), day (2) 400 µL LNPs (40 µg mRNA), day (3) 400 µL LNPs (40 µg mRNA) and harvested after 96 h. Cells treated with equal volume (200 µL, 400 µL, 400 µL) of corresponding empty-DD or empty-MC3 LNPs (without mRNA), as well as untreated cells were used as negative controls.

**Detection and quantification of hEPO mRNA in epithelial cells.** Total RNA from HTB-177 cells was isolated using miRCURY$^{TM}$ RNA isolation kit-Cell and Plant (Exiqon) according to the manufacturer's instructions. Total RNA was quantified by Qubit 2.0 fluorometer (Thermo Fisher Scientific) and the RNA quality (230/260 ratio) was assessed using NanoDrop 1000 (Thermo Fisher Scientific). Based on RNA yield, 0.25 to 1 µg of total cellular RNA was converted into cDNA using high-capacity cDNA kit (Thermo Fisher Scientific). 100 ng of cDNA was used for hEPO mRNA quantification using TaqMan probe assay (Applied Biosystems; assay ID Hs01071097_m1) on ViiA$^{™}$ 7 instrument (Thermo Fisher Scientific) according to the manufacturer's instructions. To generate the standard curve, 2 µg of pure hEPO mRNA was reverse transcribed and the resultant cDNA was serially diluted (ten-fold) to prepare seven standards (highest point: 100 ng) which were run in technical triplicate. Cellular cDNA was used for hEPO mRNA analysis whose absolute quantification was interpolated against the standard curve with minimal $R^2 > 0.975$. GAPDH (assay ID Hs02758991_g1) was used as internal control.

**Isolation of extracellular vesicles.** EVs were isolated from conditioned culture medium of LNP-treated cells and negative controls. Briefly, to remove cell debris, the cultured medium was centrifuged at 3000 × g for 15 min at 4 °C on a 4K15 centrifuge (Sigma) and the resultant supernatant was collected and ultracentrifuged at 60,000 × g for 35 min at 4 °C, followed by filtration through 0.2µm filters to obtain EVs with diameter below 200 nm. Finally, the filtered supernatant was ultracentrifuged using Optima L-100 XP ultracentrifuge with 70Ti rotor (Beckman Coulter) at 120,000 × g for 70 min at 4 °C to pellet EVs. The EV pellets were resuspended in 50–80 µl of PBS. EVs secreted after the endocytosis of LNPs were defined as endo-EVs.

**Characterization of EVs by total RNA and protein content.** EVs were quantified based on their total protein concentration and total RNA. 2 µl of EV suspension incubated together with 2 µl of M-PER Mammalian Protein Extraction Reagent (Thermo Fisher Scientific), were sonicated on an Ultrasonic cleaner (VWR) for 5 min at 54 °C to generate EV extracts. EV proteins were quantified by Qubit 2.0 fluorometer (Thermo Fisher Scientific) according to manufacturer's protocol.

Total RNA from EVs was isolated using miRCURY$^{TM}$ RNA isolation kit-Cell and Plant (Exiqon) according to the manufacturer's instructions. Total RNA was

quantified by Qubit 2.0 fluorometer (Thermo Fisher Scientific) and the RNA quality (230/260 ratio) was assessed using NanoDrop 1000 (Thermo Fisher Scientific).

**Characterization of EVs for size and concentration**. The mc3-EVs (i.e. endo-EVs isolated from MC3-LNP-treated cells) and untreated EVs were assessed for their size (nm) and concentration (particles/ml) by LM10 (Malvern Panalytical) equipped with a Hamamatsu C11440-50B/A11893-02 camera. Before the analysis, the particles were diluted 500 times in 0.1 μM filtered PBS (Sigma) to reduce the number of particles in the field of view below 180/frame. Three independent measurements (biological replicates) were performed in scatter mode. Measurement readings for each EV-sample were taken in five captures for 60 s each at 25 frames per second (fps), at adjusted camera level (10–16) and detection threshold (5–15) depending on the individual sample and manual monitoring of temperature. Blur and Max Jump Distance were set to auto. The readings, acquisition and data analysis were performed using the NanoSight Fluorescent NTA LM10 software version 3.3 (Malvern Panalytical).

**Detection of LNP-derived exogenous hEPO-mRNA in EVs**. The hEPO mRNA in endo-EVs after LNP administration and in corresponding negative controls was quantified using qPCR. Based on RNA yield, 0.25 to 1 μg of total EV-RNA was converted into cDNA using high-capacity cDNA kit (Thermo Fisher Scientific). Hundred nanograms of cDNA was used for hEPO mRNA quantification using TaqMan probe assay (Applied Biosystems; assay ID Hs01071097_m1) on ViiA™ 7 instrument (Thermo Fisher Scientific) according to the manufacturer's instructions. To generate the standard curve, 2 μg of pure hEPO mRNA was reverse transcribed and the resultant cDNA was serially diluted (ten-fold) to prepare seven standards (highest point: 100 ng) which were run in technical triplicate. EV cDNA was used for hEPO mRNA analysis whose absolute quantification was interpolated against the standard curve with minimal $R^2 > 0.975$. GAPDH (assay ID Hs02758991_g1) was used as internal control.

**Detection of EV markers and mRNA in CD63 / CD9 positive EVs**. HTB-177 cells were treated with MC3-LNPs containing 100 μg of Cy5 mRNA (Trilink) as described above. Untreated cells were included as control. After 96 h, the total EVs were isolated by UC (pre-enrichment) and quantified. After pre-enrichment, the CD63 / CD9 positive EVs were isolated by using an affinity-based method and evaluated for the presence of Cy5 mRNA by FACS. In first place, exosome-Human CD63 isolation/detection reagent for cell culture medium (Thermo Fisher Scientific) was used to immobilize the CD63$^+$ EVs to magnetic dynabeads according to manufacturer's instructions. In the binding reaction, 20 μl of beads were incubated with 25 μg or 50 μg of total mc3-EVs or total untreated EVs. As negative control, 20 μl of beads were incubated with an equivalent volume of PBS (no EVs). After the CD63$^+$ EVs immobilization, these EVs were stained with a mouse anti-human PE-CD9 antibody (BD Pharmingen™, cat. no. 555372) diluted 1:6 according to the manufacturer instructions. EVs were acquired on a BD FACSLyric system (BD Biosciences) and CD9 and Cy5 mRNA were detected and the data were analyzed using FlowJo software (TreeStar Inc.). The experiment was performed in biological duplicate. The gating strategy for beads by FACS analysis is represented in Supplementary Fig. 11a.

**Analysis of direct transfer of mRNA from LNPs into EVs**. EVs isolated from untreated cells were incubated with MC3-LNPs or DD-LNPs containing hEPO-mRNA (in PBS in absence of cells) LNPs and EVs were directly mixed and incubated (in absence of cells) at 37 °C using different proportions of LNPs and EVs. Two different proportions of EVs naïve of any prior treatment were incubated with 300 μL of DD-LNPs or MC3-LNPs (39 μg of hEPO mRNA) for 2 h in 30 mL of PBS at 37 °C. In the first setup, the proportion between EVs and LNPs was 200 μg EVs + 300 μL LNPs (39 μg of hEPO mRNA), whereas in the second condition the proportion was 50 μg EVs + 300 μL LNPs (39 μg of hEPO mRNA). After 2 h of incubation with LNP-hEPO-mRNA, the EVs were re-isolated by ultracentrifugation, total RNA from EVs was isolated and the presence of hEPO mRNA was analyzed by qPCR to evaluate whether the direct transfer of hEPO mRNA from LNPs into EVs had occurred. As negative control, equivalent volumes of DD-LNPs or MC3-LNPs were incubated in PBS without EVs and ultracentrifuged. As positive control, cells were administered with mc3-LNPs or DD-LNPs containing hEPO-mRNA and EVs were isolated (so called mc3-EVs, or dd-EVs) and mRNA was analyzed by qPCR. The experiment was performed in biological triplicate. Data are presented as percentage of hEPO mRNA detected in EVs relative to the administered amount of hEPO-mRNA delivered by LNPs to cells or hEPO-mRNA amount directly mixed with EVs. Mean values with standard deviation (SD) of replicates are shown.

**EV-mRNA protection assay**. HTB-177 cells were treated with MC3-LNPs containing 100 μg of hEPO-mRNA as described above. Untreated cells were included as control. After 96 h, the EVs were isolated and quantified. First, in order to evaluate the efficiency of the RNase A activity (Thermo Fisher Scientific), 280 ng of pure hEPO-mRNA (Trilink) was incubated with the RNase A (0.5 μg/μl) or with an equal volume of PBS at 37 °C for 20 min. Next, 200 μg of mc3-EVs were treated

with RNase A using the same conditions. As negative controls, 200 μg of mc3-EVs and 150 μg of untreated EVs were incubated in the same conditions except that RNase A was replaced by PBS. After the incubation, the total RNA from EVs was isolated using miRCURY™ RNA isolation Kit-Cell and Plant (Exiqon). The hEPO-mRNA was quantified by qPCR to assess the effect of the RNase A on hEPO-mRNA content if present outside of EVs. The experiments were performed in biological triplicate.

**Gradient UPLC for the analysis of ionizable lipids in EVs**. Fraction of EVs was used to examine the presence of LNP-derived ionizable lipids in EVs. Five to ten microlitres of each EV sample i.e. mc3-EVs and dd-EVs (obtained from MC3- and DD-LNP-treated cells, respectively) was diluted 50 times with PBS and further diluted 1 + 1 with a mixture of 2%w/v of Triton® X-100 in Tris/EDTA buffer. The samples were incubated at 37 °C for 30 min and then injected on Acquity Ultra Performance LC coupled to a Single Quad Detector, SQD (Waters, Milford). The analytical column was a Waters Acquity UPLC® CSH C18, 1.7 μm, 2.1 × 100 mm, kept at 60 °C. The flow rate was 0.50 mL/min using a mobile phase of 0.1% formic acid in water (A) and 0.1% formic acid in an equal mixture of acetonitrile and isopropylalcohol (B). A gradient run was applied where 10% B at 0.0 min was increased to 85% B at 1.0–5.0 min and kept at 85% B to 7.5 min. A washing step of 99% B at 7.6–9.5 min was included in the gradient run. Then 10% B was applied for conditioning from 9.6 min to 12.0 min. The separation between main peak of Triton X-100 and the cationic lipids was good under these conditions with retention time of Triton X-100 at 5.1 min, DLin-DMA 6.3 min and of DLin-MC3-DMA at 6.5 min. Quantification was made using external standard solutions of DLin-DMA and DLin-MC3-DMA dissolved in ethanol 99.5% for at least five different concentrations covering the expected sample concentrations with good correlation of each standard curve. The SQD was run using electrospray, positive mode and tuned using auto tune with a solution of DLin-MC3-DMA. Recording of the cationic lipids was made using Single Ion Recording (SIR) at M + 1 for each cationic lipid.

Finally, the molar ratio of ionizable lipid per hEPO mRNA nucleotides (ionizable lipid: mRNA) was determined in both EVs and LNPs. The experiments were performed in at least six biological replicates both for mc3- and dd-EVs and their corresponding LNPs.

**hEPO protein quantification**. After LNP treatment, the cell-conditioned supernatant was collected and saved for hEPO protein detection. The total cellular proteins were extracted from cell lysate using 500 μL of M-PER Mammalian Protein Extraction Reagent (Thermo Fisher Scientific) in the presence of 1% halt protease inhibitor cocktail (Thermo Fisher Scientific). Briefly, cells were gently agitated on a three-dimensional Bio-rocker for 10 min at 4 °C and centrifuged at 14,000 × g for 10 min to pellet the cell debris and the resultant supernatant (containing proteins) was transferred to a new tube. In parallel, the cultured supernatant was centrifuged at 3000 × g for 15 min at 4 °C on a 4K15 centrifuge (Sigma) to remove cell debris and EVs were isolated. To generate EV protein extracts, 2 μl of EV suspension was incubated together with 2 μl of M-PER Mammalian Protein Extraction Reagent and was sonicated on an Ultrasonic cleaner (VWR) for 5 min at 54 °C. Total proteins from all samples (conditioned supernatant, cell lysate and EV lysate) were quantified by Qubit 2.0 fluorometer (Thermo Fisher Scientific). To detect hEPO protein, the Erythropoietin ELISA Kit (STEMCELL Technologies, cat. no. 01630) was used according to the manufacturer's instructions. Fifty microliters of total proteins solution was used and hEPO protein levels were calculated according to the relative standard curve as mU/mL. The concentration was converted into fg/mL using the conversion (119mU = 1 ng) and normalized to the total number of cells.

**Effects of LNPs on cell growth, RNA, and protein content**. To determine the effect of LNPs on cellular behavior and their tolerance against LNPs treatment, the cell generation time, cellular total RNA, total amount of intracellular proteins and secreted proteins were calculated after DD- or MC3-LNP treatment period of 96 h. The effect of LNPs on EVs was also examined by quantifying EVs, total EV-RNA and protein amount of EVs against the treatment of LNPs.

The cell generation time ((G), the time (in hours) to double the population of cells) was calculated based on the difference between the number of cells at the beginning and at the end of the treatment interval (delta #cells i.e. ΔN) using the following formula:

$$G = t/n$$
$$t = \text{LNPs administration interval } (h)$$
$$n = \log(n. \text{ cells post} - \text{administration}) - \log(n. \text{ cells pre} - \text{administration})/\log 2$$

The variation of total RNA in cells and in EVs as well as total protein in EVs, total proteins in cells and in cultured supernatant were normalized to the corresponding ΔN.

**hEPO mRNA delivery to human epithelial cells via EVs**. The HTB-177 cells were seeded at a density of $5 \times 10^6$ cells/175 cm$^2$ flask and cultured in RPMI-1640 complete medium. Six hundred micrograms of mc3-EVs (700 ng hEPO mRNA)

that were isolated from MC3-LNP-treated cells and 600 µg of dd-EVs (1100 ng hEPO mRNA) isolated from DD-LNP-treated cells were dissolved in RPMI-1640 medium and different doses of these EVs were transferred to recipient cells in independent experiments over 2 days: day 1, 300 µg and day 2, 300 µg (in two time separated doses of 150 µg each after 8 h). Empty EVs (without hEPO mRNA) and EVs from untreated cells were delivered to recipient cells as control. After 48 h, cells and cultured supernatant were collected; total RNA was isolated and hEPO mRNA and hEPO protein were evaluated by qPCR and ELISA, respectively. The experiment was performed in two independent biological replicates.

**Cyanine 5 mRNA delivery to cells via endo-EVs.** One mL of DD-LNPs containing fluorescent Cy5 mRNA was delivered to HTB-177 cells in different doses (200, 400, 400 µL) with the exception that 1 mL of LNPs contained 76 µg of fluorescent Cy5 mRNA/flask (which in the case of hEPO mRNA was 100 µg/mL). Ninety-six hours post-administration of LNPs, the conditioned medium (supernatant) was harvested and used for EVs isolation. Empty DD-LNPs and untreated cells were used as controls. HTB-177 cells and immune cells such as B-cells, T-cells and monocytes purified from PBMCs were seeded at a density of $2 \times 10^5$ cells/well and cultured in 200 µl of culturing medium in 96-well round bottom plates and incubated overnight at 37 °C, 5% $CO_2$. After 24 h of stimulation of cultured cells, 78 µg of dd-EVs containing Cy5 mRNA in 25 µL PBS solution were delivered to recipient cells. As control assays, the empty dd-EVs and EVs from untreated cells were delivered to cells or cells left untreated. After 5, 24, and 48 h of EV treatment, cells were harvested and stained for surface with BV421 monoclonal antibodies (mAbs) against CD19 (B-cells), CD3 (T-cells) and CD14 (monocytes) (Becton-Dickinson Biosciences) which were diluted 1:20. Cells were acquired on a FACS-Verse (BD Biosciences), Cy5 mRNA was detected based on fluorescence in each cell type and the data were analyzed using FlowJo software (TreeStar Inc.). The gating strategy for cells by FACS analysis is represented in Supplementary Fig. 11b, c.

**Effect of pH on hEPO mRNA release from LNPs.** MC3-LNPs, containing hEPO mRNA of concentration 0.011 mg/mL were incubated in 10 mM citric acid -$Na_2HPO_4$ buffer solutions with 150 mM NaCl of various pH environments (pH 7.4, 6.6, and 5.8) at 37 °C under quiescent conditions. The total amount of mRNA was measured at time zero using 0.125 mM TritonX-100 (VWR Proteomics Grade) and 0.125 mM Sodium Dodecyl Sulfate (Sigma) in the RiboGreen Assay to be able to calculate the fraction of mRNA released. To assess the fraction of mRNA released from LNPs at various pH environments, the free amount of mRNA was analyzed with Quant-iT RiboGreen RNA Reagent Assay kit (Thermo Fisher Scientific) using a Perkin Elmer LS55 Luminescence Spectrometer (ex: 480 nm, em: 525 nm).

**In vivo transfer of hEPO mRNA via EVs and MC3-LNPs.** Experimental procedures were approved (ethical application number 83-2015) by the Regional Laboratory Animal Ethics Committee of Gothenburg, Sweden. All procedures conform to the Swedish Animal Welfare Act and regulations SJVFS 2012: 26. C57BL6/NCrl female mice ($n = 36$), 9–10 weeks of age, were purchased from Charles River Laboratory, Germany and housed in the animal facility at Astra Zeneca, Mölndal, Sweden. Mice were kept in groups of four mice per cage under standard conditions (21 °C RT, 12:12 h light-dark cycle, 45–55% air humidity) with access to a normal chow diet (R70, Lactamin AB) and water ad libitum. Environmental enrichment was provided (cartons, wooden tongue depressors, and cotton nesting pads). 100 µL of MC3-LNPs derived EVs or MC3-LNPs containing an equal dose of 1.5 µg hEPO mRNA were intravenously injected to mice ($n = 4$ per group). 100 µL of PBS were injected into control mice. Blood samples were collected from groups ($n = 4$) of mice by Vena Saphena microsampling at 2, 5, and 24 h after injection of EVs and LNPs. Blood samples collected in 35 uL EDTA-prepped capillary tubes, were centrifuged at $1700 \times g$ to collect plasma, which was kept frozen −86 °C until time for analysis. After 5, 24, and 96 h injection, groups of mice were terminated to collect organs. Mice were sedated by isoflurane anesthesia and bled from the orbital sinus, followed by cutting of the heart. Subsequently, entire organs (liver, kidney, spleen, pancreas, heart, thymus, lung and brain) were collected, snap frozen in liquid nitrogen and stored in −86 °C until time for analysis.

**Detection of human EPO protein in mouse plasma.** For the analysis of hEPO protein in the plasma after hEPO mRNA delivery via MC3-LNPs and mc3-EVs, hEPO assay was developed in-house on the Gyros platform. The capture antibody (3F6, MAIIA diagnostics) was biotinylated according to kit insert using EZ-Link Sulfo-NHS-LC-Biotin kit (Thermo Scientific). The detection antibody (7D3, MAIIA diagnostics) was Alexa 647-labelled using monoclonal antibody labeling kit (Thermo Scientific). The hEPO protein (in-house) was used to generate a standard curve in Rexxip A buffer (Gyros Protein Technologies) ranging from 12.2 pg/mL to 50 ng/mL. Mouse plasma samples were diluted 1:1 (v:v) in Rexxip A-max buffer (Gyros Protein Technologies) prior to analysis. The samples were analyzed on a Gyrolab Bioaffy 1000 CD (Gyros Protein Technologies) with Gyrolab instrument (Gyrolab xP workstation, Gyros Protein Technologies). A 5-parametric curve fitting was used for the standard curve. All standards and samples had CVs below 10%.

**Detection of human EPO protein in mouse tissues.** Total protein from organs was extracted using M-PER Mammalian Protein Extraction Reagent (Thermo Fisher Scientific) in the presence of 1% halt protease inhibitor cocktail (Thermo Fisher Scientific) following the manufacturer's instruction. Briefly, 20–70 mg of tissue were lysed in 200–350 µL of lysis buffer (depending on tissue weight) with addition of proteases inhibitors (Thermo Fisher Scientific) in the Tissue LyserII (Qiagen) for 3–5 min at the maximum speed (30 Hz) and centrifuged at $10,000 \times g$ for 15 min at 4 °C to deplete tissue debris. Resultant supernatant was used for protein quantification by Qubit 2.0 fluorometer (Thermo Fisher Scientific). Fifty microliter of total protein were analyzed for hEPO protein detection using Erythropoietin ELISA kit (STEMCELL Technologies, cat. no. 01630) according to manufacturer's instructions. The amount of hEPO protein (ng) in each organ was normalized to the relative organ weight (g).

**Cytokine analysis in mouse plasma.** After intravenous administration of MC3-LNPs and mc3-EVs, the plasma concentrations of mouse cytokines were measured by EMD Millipore's MILLIPLEX® MAP Mouse Cytokine magnetic bead kit (#MCYTOMAG-70K, Merck KGaA, Darmstadt) for the simultaneous quantification of IL-6, KC, MCP-1, RANTES, TNFα, IFNγ, IL-1β, and IP-10. The samples were first diluted 1:2 with Assay buffer and then, together with standards and QCs placed in a 96-well plate. A solution containing beads were added. The beads were magnetic microspheres each of which was coated with a specific antibody. The mixture was incubated over night at 4 °C and the reaction mixture was then incubated with Streptavidin-PE conjugate to complete the reaction on the surface of each microsphere. The plate was read on analyzer Bio Rad Luminex 200®. Each individual microsphere was identified and the result of its bioassay was quantified based on fluorescent reporter signals. The concentration was measured using Median Fluorescent Intensity data using 5-parameter logistic curve-fitting method.

**Detection of human EPO mRNA in mouse organs.** Total RNA from organs was isolated using the RNeasy kit (Qiagen) according to manufacturer's recommendations. 10–50 mg of tissue were lysed in RLT buffer (600 µL) in the Tissue LyserII (Qiagen) for 3–4 min at the maximum speed (30 Hz) and centrifuged at $10,000 \times g$ for 3 min at 20 °C to deplete tissue debris. Subsequently, the supernatant was transferred to columns and further processed. RNA was quantified via Qubit 2.0 fluorometer (Thermo Fisher Scientific) and the quality was assessed using Nano-Drop 1000 (Thermo Fisher Scientific) by measuring 260/230 ratio. Based on RNA yield, between 0.5 and 1 µg of total RNA was converted into cDNA. 100 ng of cDNA was used for hEPO mRNA quantification using TaqMan probe assay as described above. The amount of hEPO mRNA in each organ (ng) was normalized to the relative organ weight (g).

**Statistical analysis.** The statistical analysis was performed by GraphPad Prism v.7 (Graphpad Software). The in vitro data were analyzed by unpaired two-tailed Student's $t$-test, except for the effects of LNPs administration on HTB-177 growth, RNA and cell total protein amount, which were analyzed by one-way ANOVA followed by Tukey's multiple comparison test (significant $p$-value < 0.05). The hEPO content in murine plasma and organs were analyzed using unpaired two-tailed Student's $t$-test, while the levels of cytokines in mice plasma were analyzed using one-way ANOVA followed by Sidak's multiple comparisons test. The level of significance of $p$-values are indicated as follows: $*p < 0.05$, $**p < 0.01$, $***p < 0.001$, and $****p < 0.0001$.

## Data availability
The source data for main Figs. 1a–d, f–j, 2a, b, 3, 4a–p, 5, 6a–h as well as for Supplementary Figs. 1b–e, 2a–e, 3, 5e–j, 7, 8, 9a–p, 10a–h and Supplementary Table 1 are provided as a "Source Data" file. Other data are available from the corresponding author upon reasonable request.

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

## Acknowledgements

This work has been supported by grants from the Swedish research council (VR), the Swedish governmental agency for innovation systems (VINNOVA, 2017-02960), the

Swedish Foundation of Strategic Research (SSF) in the Industrial Research Centre, FoRmulaEx – Nucleotide Functional Drug Delivery (IRC15-0065) and AstraZeneca R&D Gothenburg. Moreover, we would like to express our gratitude to Professor Esbjörn Telemo, from the University of Gothenburg, for valuable discussions and comments.

## Author contributions

H.V. and L.L. conceived the study. M.M., M.N., A.P., A.A., A.C., M.N., M.H., P.S., S.J., J.L., T.K., T.J., and S.Ö. performed the experiments. M.M., M.N., A.P., A.A., A.C., M.N., M.H., P.S., S.J., M.S., J.L., T.K., I-L. M-B., T.J., P.S., S.Ö., L.L., and H.V. analyzed the data. M.N., M.M., L.L., and H.V. wrote the paper. All authors reviewed, edited, and approved the final version of the paper.

## Additional information

**Competing interests:** The authors declare no competing interests

**Peer Review Information** *Nature Communications* thanks Harald Köfeler and other, anonymous, reviewers for their contribution to the peer review of this work. Peer reviewer reports are available.

