## [Peer Review File · Nature Communications]

Reviewers' comments:

Reviewer #1 (Remarks to the Author):

The article by Maugeri et al describes the use of lipid nanoparticles (LNP) containing encapsulated mRNA as a tool for delivery of the mRNA in target cells, leading to subsequent release of EVs containing also the mRNA and some lipids from the LNPs, which are also able to transfer the mRNA to target cells. The EVs from LNP-treated cells elicit less inflammatory responses when injected in animals than the original LNPs themselves, and the authors propose to use them as safer delivery vehicles for in vivo therapy.

Experiments are performed and controlled in an appropriate manner. The observations are interesting for clinical applications, although improvements of the use of LNP-EVs rather than LNPs themselves is not very striking: they observe similar or lower and no prolonged expression of the protein (not detected anymore at 24h), and the inflammatory reaction induced by LNPs and not by EVs is also transient (not detected at 24h). Thus, whether LNP-EVs are really worth evaluating further for therapeutic applications will require additional experiments with therapeutic outcomes (of course, for a follow-up article and not for the current article).

for this part of the results, the following minor changes in presentation should be implemented: Replace bar graphs by dot plots (or include dots as in fig1H-I) in figS1, fig1A-D, fig1F, as recommended for all publications, to show the distribution of individual biological replicates (see: Weissgerber et al. (2015) Beyond Bar and Line Graphs: Time for a New Data Presentation Paradigm. PLoS Biol 13(4): e1002128.)

Figure 2 is neither informative nor consistent with the rest of the article: the authors use DD-EVs here, whereas their other experiments suggest that MC3-EVs are more efficient at delivery, and the read-out, of Cy5 fluorescence in target cells, does not give any indication on the efficiency of delivery of a functional mRNA, translated into protein. This figure should be deleted.

Figures S6 and S7 should be shown as main figure, side by side with figures 3 and 4, specifying the actual amount of mRNA injected in both conditions, to show the actual comparison of LNPs and LNP-EVs. The conclusion is that EVs are not more efficient than LNPs, for expression of the protein encoded by the mRNA, and that the target tissues are similar but with an additional one (kidney) when using LNPs. This conclusion is confirmed in fig5, when the same amount of mRNA, delivered either in LNP-EVs or in LNPs is compared.

Another message of this article is more on a basic science point of view: the authors claim a link between endosomal escape of the LNPs and subsequent release in EVs of endosomal origin. However, since they do not perform any cell biology experiments to demonstrate the link with endocytosis of LNPs, this claim, and the model of figure 7 are not properly supported. In fact, this model proposes first endosomal escape of mRNA from LNPs into cytosol (presumably because of the different charges of lipids in LNPs and the endosome membrane) then reinsertion of mRNA into ILVs of MVBs, whereas LNP sticking to EVs inside MVBs would seem easier to achieve! Such an interaction between LNPs and EVs could even happen outside cells on EVs released from MVBs or from the PM.

Experimentally, the authors isolate EVs by a classical differential ultracentrifugation protocol that co-isolates EVs of any intracellular origin, endosomal or not. In addition, LNPs have apparently a size similar to that of EVs (84-88nm diameter), suggesting that they may co-pellet with EVs. The ultracentrifugation isolation protocol (like all crude isolation processes) may also co-precipitate a proportion of the LNPs with EVs released by the cells: the different proportion of ionizable lipids in LNPs and EVs could result from a small proportion of LNPs in the EV preps, rather than actual incorporation of the LNPs' lipids in EVs. The authors only improperly answer this concern by mixing LNPs with isolated EVs, re-isolating EVs, and finding LNP-derived lipids in them. In fact, LNPs fed to cells are exposed to a culture medium that could change their surface and aggregation properties, and lead to their acquired ability to pellet during the EV isolation process. The proper experiment would thus be to mix isolated EVs in culture medium (or, more straightforward, cell culture conditioned medium containing EVs) with or not LNPs for the time of culture used in vitro

(ie total 48h), before performing the EV isolation process, and compare the resulting EV preps side by side.

In addition, to separate more precisely LNPs from EVs (if they are not covalently bound to each other) the EVs could be immunoprecipitated with an antibody to a surface protein (a tetraspanin, or any other), and their ionizable lipid assessed. The EPO mRNA content should also be measured after or not exposure to RNase to determine whether RNA is protected differently by LNPs, EVs, and LNP-EVs.

In general, beside the endosomal origin or not of the EVs analysed here, the article does not provide enough demonstration that EVs are carrying the RNA from LNPs because no characterization of these EVs is provided. Electron microscopy images of LNPs, LNP-EVs and EVs could be useful here. And whether these EVs come from endosomes, or from another part of the cell (plasma membrane) is not addressed at all. Thus, the use of a term like endo-EVs by the authors is really not justified here.

Since unraveling the cell biology process of LNP association with EVs and/or transfer of LNP-RNA into EVs is not really necessary for the subsequent use as delivery vehicle, the authors could choose to delete any claims on endosomal escape and EV-mediated delivery, but a proper characterization of EVs would anyway be necessary for future therapeutic applications.

Reviewer #2 (Remarks to the Author):

The authors have described that mRNA delivered by LNPs can be incorporated into endo-EVs, which can then be isolated and used for delivery.

The authors should be commended on several aspects of the paper.

- (1) The hypothesis is original. I have not heard of this idea in the field previously.
- (2) The authors did a nice job characterizing the cellular response to the LNPs in figure 1.
- (3) The authors have included appropriate, unbiased citations throughout the paper.

However, there are several aspects that could strengthen the paper.

(1) Page 5. The sentence '... first to show ... that the endosomal escape of exogenously delivered mRNA is dependent on the molar ratios...' is too broad. There are several papers from the Anderson lab at MIT that show molar ratios improve mRNA delivery in vivo. I would rewrite this sentence.

(2) Page 6. I would include the dosing and timepoints your description of figures 1. It is possible to find that information in the supplement, but it is much easier for the reader to interpret the data if the experiment is detailed more in the text than it is now.

(3) Figure 1 needs to be remade. The font is far too small. The same goes for Figure 7C.

(4) It is hard to understand how the MC3 LNPs could be simultaneously delivering more mRNA into the cytoplasm (leading to more delivery in Figure 1B,C) AND also be recycled out more in the EVs (Fig. 1F).

(5) Page 8. The argument about the molar ratio of ionizable lipids in the EVs is difficult to follow. In addition, the related figure panel in 1H is a bit confusing... the LNP concentration (in μM) went down, but the concentration of hEPO (in μM) either stayed the same or even increased a bit?

(6) What molar percentage of the EV is made up of the ionizable lipids? I apologize if I missed this in the paper, but I couldn't find it clearly.

(7) Why are the Y axis units for S3A-C and Figure 1B,C different?

(8) Page 8. The statement that '... cells are more tolerant to EV-based delivery...' should be removed. In vitro data from a cell line are not sufficient to make this blanket statement.

(9) Given that an important part of this paper is comparing the LNPs to the EVs derived from the LNPs, figure 2 (and all subsequent figures) should have LNPs as a control group, on the same figure.

(10) Finally, related to comment 7, given that in all drugs, therapeutic windows dictate toxicity, it is not appropriate to compare toxicology of LNPs to EVs at the same dose if - at that dose - the LNPs are delivering more protein than the EVs. The appropriate comparison is dose X for LNPs and dose Y for EVs, where X and Y result in equal amounts of protein production.

Reviewer #3 (Remarks to the Author):

Regarding the lipid analysis section of the manuscript several flaws are evident in the workflow. First of all it is not recommended to use Triton X-100 when working with lipids. This is due to the highly suppressive nature of this agent in the electrospray ionization process of lipids. Furthermore there is clearly room for improvement when it comes to the selectivity of instrumentation. A state of the art setting in respect to selectivity would at least include either high mass resolution or MS/MS capabilities, if not even both of them. Just retention times and the corresponding nominal masses of molecular adduct ions can be regarded rather weak evidence. So the recommendation here would be to use either a triple quadrupole or an ESI-TOF / Exactive instead of a single quad for enhanced identification certainty. The weakest part of the workflow is the quantitation part. To come up with quantitative numbers of lipids at least a one point calibration with a suitable internal standard would be needed, a calibration curve for each quantified lipid would be even better. Due to matrix effects (differential ion suppression effects) between external calibration samples and unknown samples and further even between unknown samples (intra batch) and between analytes within one sample (intra sample), the performed external quantitation can be deemed highly risky in ESI-MS, resulting eventually in massively crunched numbers for the quantity of individual lipids. As an example, just the use of Triton X-100 in the unknown samples but not in the calibration samples can lead to a huge shift of numbers.

Response to reviewer comments

We are very grateful for the professional comments and constructive critiques made by referees. Under their guidance, we managed to have a much-improved manuscript by conducting new experiments and have included new results in the manuscript. We have added the corresponding methods, as well as the raw data with Source data files. In addition, we have also changed / presented figures as the reviewers had proposed.

All the updates can be perceived in the point-by-point reply below, and the corresponding tracked changes added in the main manuscript file.

Reviewer #1 (Remarks to the Author):

The article by Maugeri et al describes the use of lipid nanoparticles (LNP) containing encapsulated mRNA as a tool for delivery of the mRNA in target cells, leading to subsequent release of EVs containing also the mRNA and some lipids from the LNPs, which are also able to transfer the mRNA to target cells. The EVs from LNP-treated cells elicit less inflammatory responses when injected in animals than the original LNPs themselves, and the authors propose to use them as safer delivery vehicles for in vivo therapy. Experiments are performed and controlled in an appropriate manner. The observations are interesting for clinical applications, although improvements of the use of LNP-EVs rather than LNPs themselves is not very striking: they observe similar or lower and no prolonged expression of the protein (not detected anymore at 24h), and the inflammatory reaction induced by LNPs and not by EVs is also transient (not detected at 24h). Thus, whether LNP-EVs are really worth evaluating further for therapeutic applications will require additional experiments with therapeutic outcomes (of course, for a follow-up article and not for the current article).

Response

Dear reviewer,

We are very much thankful for the constructive critique and comments. The point-by-point answers are provided below.

1. For this part of the results, the following minor changes in presentation should be implemented: Replace bar graphs by dot plots (or include dots as in fig1H-I) in figS1, fig1A-D, fig1F, as recommended for all publications, to show the distribution of individual biological replicates (see: Weissgerber et al. (2015) Beyond Bar and Line Graphs: Time for a New Data Presentation Paradigm. PLoS Biol 13(4): e1002128).

Response:

As suggested by reviewer, we have replaced the bar graphs by dot plots in Fig1A-D, and FigS1.

2. Figure 2 is neither informative nor consistent with the rest of the article: the authors use DD-EVs here, whereas their other experiments suggest that MC3-EVs are more efficient at delivery, and the read-out, of Cy5 fluorescence in target cells, does not give any indication on the efficiency of delivery of a functional mRNA, translated into protein. This figure should be deleted.

Response:

We agree with the reviewer, and we could have removed these results, but this is the first time to show deliver foreign mRNA to human blood cells (healthy donors). In addition, for example, it has previously been shown that it is difficult to introduce any genetic materials (RNA/ DNA) into B-cells. We thought that it would be of value to show that EVs are capable of delivering foreign mRNA to human B-cells, T-cells, and monocytes. Moreover, as per editor's comment we have been suggested to keep the Figure 2 as main figure.

3. Figures S6 and S7 should be shown as main figure, side by side with figures 3 and 4, specifying the actual amount of mRNA injected in both conditions, to show the actual comparison of LNPs and LNP-EVs. The conclusion is that EVs are not more efficient than LNPs, for expression of the protein encoded by the mRNA, and that the target tissues are similar but with an additional one (kidney) when using LNPs. This conclusion is confirmed in fig5, when the same amount of mRNA, delivered either in LNP-EVs or in LNPs is compared.

Response:

We totally agree with the reviewer that LNPs are more efficient in current study and it has also been previously shown that LNPs are capable of delivering siRNA and mRNA to cells both *in vitro* and *in vivo*, and currently being applied in phase 2 clinical studies. Reviewer refers here the figure 5 for the efficiency of LNPs than EVs, for the delivery of the same amount of hEPO mRNA to different organs of mice. LNPs are more efficient in current study, however, in this paper we would emphasis the value of EVs as new or emerging biological delivery vehicle, and we are certain that these EVs can be improved, for example, for the adequate amount of mRNA to be inserted into EVs, their improved delivery efficiency, and to make them more cell/ organ specific. We consider that it is too early to compare EVs with LNPs, particularly when for the first time these EVs are used to deliver foreign mRNA to cells. However, we believe that certainly these EVs can be improved for this purpose.

4. Another message of this article is more on a basic science point of view: the authors claim a link between endosomal escape of the LNPs and subsequent release in EVs of endosomal origin. However, since they do not perform any cell biology experiments to demonstrate the link with endocytosis of LNPs, this claim and the model of figure 7 are not properly supported. In fact, this model proposes first endosomal escape of mRNA from LNPs into cytosol (presumably because of

the different charges of lipids in LNPs and the endosome membrane) then reinsertion of mRNA into ILVs of MVBs, whereas LNP sticking to EVs inside MVBs would seem easier to achieve! Such an interaction between LNPs and EVs could even happen outside cells on EVs released from MVBs or from the PM.

Response:

We thank the reviewer for this comment.

The link between LNP endocytosis and endosomes has already been established. Marino Zerial Lab has shown that LNPs are internalized/endocytosed by recipient cells and they co-localize to different compartments including endosomes within early hour of uptake (Ref. #5 in our manuscript). Upon internalization of LNPs, cargo is sequentially transported through early endosomes, late endosomes and lysosomes (so named biogenesis and maturation of LNP-containing organelles). Additionally, Anders Wittrup and colleagues have also provided the link between LNP endocytosis and endosomes (Ref. #4 in our manuscript). It is widely accepted by several evidence that what has translocated to endosomes can be package into intraluminal vesicles (ILVs) of multivesicular bodies (MVBs), and secreted as exosomes.

Regarding the endosomal escape and model in Figure 7, in this work, and in previous work, it has been shown that a part of the LNP-RNAs do endosomal escape and translocated in the cytoplasm of recipient cells (could be called “functional RNA” since, for example, the LNP-mRNA gives rise to the production of a new protein in the cytoplasm of cells).

Since some of the LNP material (mRNA and ionizable lipids) were detected in EVs, it means that these materials (mRNA and ionizable lipids) have incorporated within the EVs during their biosynthesis, probably when intra-luminal vesicles (ILVs) of MVBs are formed. If LNPs transfer some of their material (mRNA and ionizable lipids) to ILVs - within MVB/ late endosomes - then the ratio of LNP mRNA to ionizable lipids would be the same (1 mole of mRNA nucleotides per 3 moles of ionizable lipids). But we find that EVs contain 1 mol mRNA / 1 mole lipids.

Therefore, we assume that LNP mRNA and lipids end up in ILVs (inward invagination of endosomal membrane) after endosomal escape, but not LNP sticking within MVB. Then, there was only one way to explain endosomal escape - based on our results (1:1 molar ratio), that if LNP-mRNA is in neutrally charged (1 mol mRNA / 1 mole ionizable lipids) then mRNA can pass across the endosome membrane and end up in the cytoplasm. If the molar ratio is anything other than 1: 1, the mRNA cannot pass across the endosome membrane (to make endosomal escape) to be translocated into the cytoplasm of cells.

With these reasoning, we have presented a plausible mechanism of endosomal escape of LNP-mRNA. But, we are also aware that the future / other researchers can demonstrate other models, than this, for endosomal escape.

5. Experimentally, the authors isolate EVs by a classical differential ultracentrifugation protocol that co-isolates EVs of any intracellular origin, endosomal or not. In addition, LNPs have apparently a size similar to that of EVs (84-88nm diameter), suggesting that they may co-pellet with EVs. The ultracentrifugation isolation protocol (like all crude isolation processes) may also co-precipitate a proportion of the LNPs with EVs released by the cells: the different proportion of ionizable lipids in LNPs and EVs could result from a small proportion of LNPs in the EV preps, rather than actual incorporation of the LNPs' lipids in EVs. The authors only improperly answer this concern by mixing LNPs with isolated EVs, re-isolating EVs, and finding LNP-derived lipids in them. In fact, LNPs fed to cells are exposed to a culture medium that could change their surface and aggregation properties, and lead to their acquired ability to pellet during the EV isolation process. The proper experiment would thus be to mix isolated EVs in culture medium (or, more straightforward, cell culture conditioned medium containing EVs) with or not LNPs for the time of culture used in vitro (i.e. total 48h), before performing the EV isolation process, and compare the resulting EV preps side by side.

Response:

We agree with the reviewer about the fact that differential ultracentrifugation (UC) co-isolates EVs of any intracellular origin, endosomal (exosomes) or plasma membrane derived (microvesicles) etc. Now in our manuscript we have referred UC isolated EVs as both exosomes and microvesicles (**Box 1**).

Regarding, co-pelleting of LNPs with EVs using UC isolation protocol, we agree with the reviewer that there is possibility for co-precipitation. However, based on evidence that 95% of LNPs are inside the cell cytoplasm already within half hour, meaning that outside LNPs is very minute. Marino Zerial Lab has shown that LNPs are endocytosed with in first hour – reference #5, *Nat. Biotechnology* (while other groups show with in half hour 95% of LNPs are within the cells). Additionally, Anders Wittrup and colleagues have shown that RNA release from maturing endosomes occurred invariably within ~5–15 min of endocytosis – Ref. #4, *Nat. Biotechnology*. These studies show that that most of the LNPs are taken up (endocytosed) by cells, and the process of LNP uptake is as fast as in minutes at which point the secretion of EVs from treated cells has not achieved. This indicates out the cells there are less than 5% LNPs and at this point EVs are not secreted, thus the chances of LNPs interaction with EVs are less for co-precipitation factor. Additionally, the mRNA inside the cell is translated into protein (Fig 1b and 1c), which means LNPs are inside the cells. Therefore, in the light of ref. #5 and Figure #1,

and Suppl. figure S2B, we believe that different proportion of ionizable lipids in LNPs and EVs is due to endocytic processing of LNPs rather than acquiring a small proportion of LNPs in the EV preps from outside out cell (medium). Additionally, we have observed that during the first run of ultracentrifugation (60,000 x g for 35), LNPs, which consists of a lot of fat, float on the surface and a big white pellet is achieved, which is not observed in untreated samples. We are convinced that this white material are lipid moieties which are not observed in untreated samples. We discard these while moieties, and process the resultant supernatant for next run of UC 120,000 x g for 70 min following the filtration. However, the chances between outside interactions cannot be excluded, though less in percentage, we are uncertain about this small proportion outside of cell. But what is sure is that mRNA-loaded LNPs are made up of [Ionizable Lipid:Cholesterol:DSPE:PEG], and there are three molecules of ionizable lipid per each mRNA nucleotide i.e. the ratio between ionizable lipid and mRNA

nucleotides is 3:1. . They are held so firmly that outside of cells or in free medium regardless LNPs interact with EVs or not, the ionizable part of LNPs remains intact. What it means is, if direct fusion of LNPs with EVs had occurred, the ratio between ionizable lipid and mRNA nucleotide in EVs would have been the same (i.e. 3:1). As reviewer asked “the different proportion of ionizable lipids in LNPs and EVs could result from a small proportion of LNPs in the EV preps, rather than actual incorporation of the LNPs’ lipids in EVs”. Based on LNPs composition, we argue that even a small proportion of LNPs fuses with EV preps, still this small proportion should carry the intact ionizable lipid part with same ratio of ionizable lipid per mRNA nucleotide (3:1). However, in EVs, the calculated ratio between ionizable lipid and mRNA nucleotide was 1:1, which means that ionizable part of LNPs has been processed in the cells, where each ionizable is detached from mRNA nucleotide, which is not possible outside the cell.

Finally, regarding mixing the isolated EVs in culture medium with or without LNPs to observe whether there is direct transfer of LNPs to EVs in the absence of cells. We isolated EV isolation and analyzed for the presence of hEPO-mRNA in EVs, but there was no hEPO-mRNA in EVs that were mixed with LNPs (**Suppl. Fig. S2B**). This correlation is not the same as was observed when LNPs were administered to cells and hEPO-mRNA was observed in EVs. While mixing LNPs with EVs (in absence of cells) LNPs might have fused/aggregated with EVs or degraded in free medium and released their RNA. What has happened is not clear, however, what we see is that in the absence of cells, LNPs were unable to incorporate hEPO-mRNA into EVs.

In addition, we have conducted new experiments (immunoprecipitation and RNase treatment of EVs) (Supplementary Figure S2A and S2C), which show that EVs from LNP-treated cells (a) contain LNP-

mRNA located on the inside of these EVs (Supplementary Figure S2C) and (b) these EVs are positive for the exosome markers CD63 and CD9 as well as LNP-mRNA ((Supplementary Figure S2A).

6. In addition, to separate more precisely LNPs from EVs (if they are not covalently bound to each other) the EVs could be immunoprecipitated with an antibody to a surface protein (a tetraspanin, or any other), and their ionizable lipid assessed. The EPO mRNA content should also be measured after or not exposure to RNase to determine whether RNA is protected differently by LNPs, EVs, and LNP-EVs.

Response

Thank you very much for the comment.

We have conducted new experiments (immunoprecipitation and RNase treatment of EVs) and included these results in the manuscript: Supplementary Figure S2A (Immunoprecipitation) and Supplementary Figure S2C (RNase Treatment).

We have immunoprecipitated EVs with CD63 and CD9 antibodies (EV markers), which compared to UC is specific method to fetch EVs. We have performed FACS analysis to show that CD63 and CD9 positive EVs are also positive for administered LNP-mRNA. Our results (Suppl. Fig. S2A) show that EVs positive for CD63 and CD9 are also positive for LNP-mRNA.

The RNase treatment of EVs (EVs derived from LNP-treated cells) clearly demonstrated that LNP-mRNA is located on the inside of these EVs (Supplementary Figure S2C).

7. In general, beside the endosomal origin or not of the EVs analysed here, the article does not provide enough demonstration that EVs are carrying the RNA from LNPs because no characterization of these EVs is provided. Electron microscopy images of LNPs, LNP-EVs and EVs could be useful here. And whether these EVs come from endosomes, or from another part of the cell (plasma membrane) is not addressed at all. Thus, the use of a term like endo-EVs by the authors is really not justified here.

Response

This is important point, using quantitative fluorescence imaging and electron microscopy, it has already shown that endocytosed LNPs are translocate to endosomal compartments (as described above in response of point 4), and endosomal compartments give rise to EVs. In our work (Figure 1B and 1C), we observe the production of EPO-protein after the transfer of EPO-mRNA to cells via LNPs. It means that a part of the LNP-EPO-mRNA do endosomal escape and end up in the cytoplasm of recipient cells ("functional RNA"). In addition, we could also detect that some of the LNP material (mRNA and ionizable lipids) can be detected in EVs. However, unlike LNPs that contain 1: 3 mole ratio

of mRNA: ionizable lipids, we could detect that EVs, from LNP-treated cells, contain 1 mol mRNA / 1 mole ionizable lipids.

We reasoned that these EVs have received a part of their compositions from endocytosis of LNP-particles. Therefore, in some way, we needed to distinguish these EVs – containing LNP-mRNA and ionizable lipids - from a more common EVs that could imply EVs including plasma membrane vesicles and microvesicles. Although, we have not tracked the LNPs by microscopy, but we refer the elegant study by Marino Zerial Lab which has shown LNP-RNA inside endosomes.

We agree with the reviewer that these EVs could also be originated from the plasma membrane (so called microvesicles) and the term Endo-EVs is not justified. Therefore, we have redefined the term endo-EVs (Now Endo-EVs refer all those EVs secreted after the endocytosis of LNPs, and are redefined on **page 2, page4, and Box 1 and other places**). Regardless of the origin of EVs and chances of aggregation with LNPs, our main scope is to show these EVs (vesicles from LNP-treated cells) contain 1: 1 mole ratio of mRNA: ionizable lipids, and these EVs can transport these molecules *in vitro* and *in vivo* to cells. No doubt, the more improved methods are required, at least at this stage we show initial data that we perceive, is a starting point for future studies.

8. Since unraveling the cell biology process of LNP association with EVs and/or transfer of LNP-RNA into EVs is not really necessary for the subsequent use as delivery vehicle, the authors could choose to delete any claims on endosomal escape and EV-mediated delivery, but a proper characterization of EVs would anyway be necessary for future therapeutic applications.

Response

We agree with the reviewer that more characterization of these EVs (extracellular vesicles from LNP-treated cells) is needed to be done. In this work, we have isolated these EVs, using multiple ultracentrifugation and filtration, and were able to detect some of the LNP-materials (LNP mRNA and lipid) in these EVs. In order to characterize these EVs, we have performed immunoprecipitation, RNase treatment, size determination (NTA analysis) and carefully determined their mRNA and lipids derived from LNPs (New data is presented in Suppl. Fig S2, S3, and S4).

We agree with the reviewer that for future therapeutic applications of EVs as delivery vehicles, we need to better characterize them (EVs secreted by cells treated with LNPs).

Reviewer #2 (Remarks to the Author):

The authors have described that mRNA delivered by LNPs can be incorporated into endo-EVs, which can then be isolated and used for delivery.

The authors should be commended on several aspects of the paper.

- (1) The hypothesis is original. I have not heard of this idea in the field previously.
- (2) The authors did a nice job characterizing the cellular response to the LNPs in figure 1.
- (3) The authors have included appropriate, unbiased citations throughout the paper.

However, there are several aspects that could strengthen the paper.

Response

We are very much thankful for the reviewer for evaluating the originality and novelty of the current study.

The point-by-point answers are provided below.

- (1) Page 5. The sentence '... first to show ... that the endosomal escape of exogenously delivered mRNA is dependent on the molar ratios...' is too broad. There are several papers from the Anderson lab at MIT that show molar ratios improve mRNA delivery in vivo. I would rewrite this sentence.

Response

Thank you for pointing out this. Based on this, we have modified the sentence (page 4 – tracked changes).

- (2) Page 6. I would include the dosing and time points your description of figures 1. It is possible to find that information in the supplement, but it is much easier for the reader to interpret the data if the experiment is detailed more in the text than it is now.

Response

Thank you for the suggestion. We have included the time points in the description of Figure 1 at page #6 – tracked changes.

- (3) Figure 1 needs to be remade. The font is far too small. The same goes for Figure 7C.

Response

Based on reviewer's comment we have increased the font size of Figure 1 and Figure 7.

- (4) It is hard to understand how the MC3 LNPs could be simultaneously delivering more mRNA into the cytoplasm (leading to more delivery in Figure 1B,C) AND also be recycled out more in the EVs (Fig. 1F).

Response

Thank you for raising this point. This is because MC3-LNPs (i.e. DLin-MC3-DMA) are more efficient and improved LNPs that have not only proven for elevated uptake (and thus deliver more RNA to the cytoplasm), but also good at endosomal escape compared to DD-LNPs (i.e. DLin-DMA).

(5) Page 8. The argument about the molar ratio of ionizable lipids in the EVs is difficult to follow. In addition, the related figure panel in 1H is a bit confusing... the LNP concentration (in μM) went down, but the concentration of hEPO (in μM) either stayed the same or even increased a bit?

Response

We have performed this experiment 13 times and we have obtained the same result that the molar ratio of hEPO mRNA to ionizable lipids is 1: 1 in EVs. *We have received a varying amount of mRNA in EVs in different experiments, but the ratio between LNP-mRNA and ionizable lipids has been the same (1: 1 molar ratio in EVs).* It is different from what was present in originally manufactured LNPs (the molar ratio in LNPs is 1: 3). Therefore, in this work, we claim that LNP-mRNA first makes endosomal escape and then ends in intraluminal vesicles (ILVs) of MVBs. We reason that if the LNP-mRNA enters in ILVs, when LNPs are in endosomes, then the ratio of mRNA and ionizable lipids should be the same as LNPs, *i.e.* 1: 3 mole.

(6) What molar percentage of the EV is made up of the ionizable lipids? I apologize if I missed this in the paper, but I couldn't find it clearly.

Response

Unfortunately, we cannot answer the question today, how much of the total components of EVs consists of ionizable lipids. It is an interesting question and there might be ways to determine it, but unfortunately, we cannot answer the question today.

(7) Why are the Y axis units for S3A-C and Figure 1B,C different?

Response

We thank the reviewer for pointing out this discrepancy. In fact, we want to show the sum of hEPO protein produced (intracellular + extracellular). Therefore, we calculate "mole" amount of hEPO-protein.

(8) Page 8. The statement that '... cells are more tolerant to EV-based delivery...' should be removed. In vitro data from a cell line are not sufficient to make this blanket statement.

Response

Based on reviewer comment, we have deleted this statement (page 10 of current revision- tracked changes).

(9) Given that an important part of this paper is comparing the LNPs to the EVs derived from the LNPs, figure 2 (and all subsequent figures) should have LNPs as a control group, on the same figure.

Response

We thank the reviewer for this suggestion. It has been previously shown that LNPs are capable of delivering siRNA and mRNA to cells both *in vitro* and *in vivo*, and currently being applied in phase 2 clinical studies. However, in this paper we would emphasize the value of EVs as new or emerging biological delivery vehicles that is why we focus on EV part. We are certain that these EVs can be improved, for example, for the adequate amount of mRNA to be inserted into EVs, their improved delivery efficiency, and to make them more cell/ organ specific. We consider that it is too early to compare EVs with LNPs, particularly when for the first time these EVs are used to deliver foreign mRNA to cells. However, we believe that certainly these EVs can be improved for this purpose.

(10) Finally, related to comment 7, given that in all drugs, therapeutic windows dictate toxicity, it is not appropriate to compare toxicology of LNPs to EVs at the same dose if - at that dose - the LNPs are delivering more protein than the EVs. The appropriate comparison is dose X for LNPs and dose Y for EVs, where X and Y result in equal amounts of protein production.

Response

We agree with the reviews. In addition, EVs is a biological vehicle, containing a lot of other RNAs and proteins and lipid, and LNPs is a chemical product. We also think they should not be compared to each other, they can fit for different purposes. LNPs are highly effective RNA providers especially *in vitro*, even to odd cell types or stem cells.

The only way we could compare these two vehicles (LNPs and EVs) was to deliver equal amount of mRNA *in vivo* via either LNPs or EVs. It is true that if EVs would produce equal amount of EPO protein in mice, would they also induce as much immune response as LNPs. We are convinced that we need to do more experiments, e.g. to isolate only the population of EVs that contain LNP-mRNA, and then transfer only these EVs to mice, in comparison with LNPs.

Reviewer #3 (Remarks to the Author):

Regarding the lipid analysis section of the manuscript several flaws are evident in the workflow. First of all it is not recommended to use Triton X-100 when working with lipids. This is due to the highly suppressive nature of this agent in the electrospray ionization process of lipids. Furthermore there is clearly room for improvement when it comes to the selectivity of instrumentation. A state of the art setting in respect to selectivity would at least include either high mass resolution or MS/MS capabilities, if not even both of them. Just retention times and the corresponding nominal masses of molecular adduct ions can be regarded rather weak evidence. So the recommendation here would be

to use either a triple quadrupole or an ESI-TOF / Exactive instead of a single quad for enhanced identification certainty. The weakest part of the workflow is the quantitation part. To come up with quantitative numbers of lipids at least a one point calibration with a suitable internal standard would be needed, a calibration curve for each quantified lipid would be even better. Due to matrix effects (differential ion suppression effects) between external calibration samples and unknown samples and further even between unknown samples (intra batch) and between analytes within one sample (intra sample), the performed external quantitation can be deemed highly risky in ESI-MS, resulting eventually in massively crunched numbers for the quantity of individual lipids. As an example, just the use of Triton X-100 in the unknown samples but not in the calibration samples can lead to a huge shift of numbers.

Response

Dear Reviewer,

Thank you for your comments, you point out an important aspect on how we have quantified the ionizable lipid component in the different EV samples. Below we show results from additional experiments to show how the presence of Triton X-100 and “untreated” EVs influence quantification using UPLC-MS. The results are illustrated for the DLin-MC3-DMA lipid, but similar results are obtained for the other lipid. The data is generated using the same UPLC gradient, column and mass spectrometer as used in the generation of the manuscript data.

We agree with your comment that Triton X-100 has a negative influence using MS as detection. Due to the relative high MS sensitivity for ionizable lipids at acidic pH and the high concentrations of each ionizable lipid in the EV samples, the samples were diluted totally 100 times before injection. **Suppl. figure S4** illustrate, for the same concentration range observed for such diluted samples in the manuscript, that the ion suppression effect is small for this setup of chromatographic method/mass spectrometer. This is likely due to the different retention times for the Triton-X100 and ionizable lipid components. In these experiments, the lipid has been dissolved in either ethanol, 1% (w/w) Triton X-100 or in 1% (w/w) Triton-X100 spiked with a representative sample of “untreated” EVs, and injected on the UPLC-MS system. The response for the Triton X-100 (with and without “untreated” EVs) samples is lower compared to the samples dissolved in pure ethanol, but we estimate the error in quantification to be less than 10%.

HPLC-MS analysis of DLin-MC3-DMA samples

Suppl. figure #4. UPLC-MS analysis of DLin-MC3-DMA samples prepared in ethanol (**red**), in 1% (w/w) Triton X-100 (**green**) and in 1% (w/w) Triton X-100 with added fixed amount “untreated” EVs (**blue**).

Considering the likely variation in the sample preparations (for mRNA and lipid quantifications) and the actual qPCR analysis (mRNA) and UPLC-MS (lipids) it is our opinion that the observed error does not influence the main conclusion that mRNA and the ionizable lipid are co-transported as a complex stoichiometric salt (1:1) into the secreted EVs. Furthermore, it should be noted that we observe a very strong correlation between nucleotide concentration measured using qPCR and lipid concentration determined using UPLC-MS for the different EV samples, see Figure 1J of the original manuscript.

For further clarifications, we have added the following statement in our manuscript, page 23 “Quantification was made using external standard solutions of DLin-DMA and DLin-MC3-DMA dissolved in ethanol 99.5% for at least 5 different concentrations covering the expected sample concentrations with good correlation of each standard curve”.

Finally, we have updated the methods for this section, on page 23 of our manuscript – tracked changes.

Reviewers' comments:

Reviewer #2 (Remarks to the Author):

I have re-read the paper, and have also considered the arguments made by reviewer 1. In general, I do not believe the authors have responded effectively enough to Reviewer 1's critiques or my own to justify publication. I still share Reviewer 1's concerns about how they are isolating the EVs.

A few examples are described below...

Rebuttal to Reviewer 1, point 2. In Figure 2, the authors write that they show delivery to B cells. It is unclear if they do. If they are quantifying GFP, then they do. If (more likely) they are reading out Cy5, they don't... they only have uptake in the cells. Moreover, this is not the first time B cells have been targeted.

Rebuttal to me, point (4). I still do not see how - from a conservation of mass perspective - their results can be explained. I understand that the potential efficacy will differ, but it still seems strange that more RNA can go into the cell AND more RNA can go out of the cell at the same time.

If the paper is published, I think the authors should remove all statements speculating these vehicles > LNPs. In their rebuttal, they say to reviewer 1 (and to me, separately) that it's too early to compare EVs to LNPs. However, in their abstract they write 'Compared with LNPs, EVs elicit significantly less inflammatory response, and may function as a safer vehicle for exogenous mRNA delivery.' That (big big big) statement is not substantiated by (1) their own rebuttal or (2) their data.

Reviewer #3 (Remarks to the Author):

The reviewers concerns are now sufficiently addressed. The limitations of the quantification are now clearly stated and do not conflict with the core message of this experiment.

Harald C. Köfeler

We thank the reviewers for their valuable time to review their comments. In this letter we would like to *(a)* stress the importance of this paper further, regarding delivery of functional mRNA to cells using both biological vehicles (e.g. EVs) and chemically formulated vehicles (e.g. LNPs), and *(b)* answer to the comments given by the Reviewer #2 (blue colored text), followed by our responses below each comment (black colored text).

Importance of this paper:

In this work, we have studied the delivery of functional mRNA to cells *in vitro* and *in vivo* via both extracellular vesicles (EVs) and lipid nanoparticles (LNPs).

The LNPs containing ionizable amino lipids, currently represent the most advanced platform for RNA delivery, which have now advanced into human clinical trials, and their mRNA delivery safety profiles have been evaluated in human and non-human primates. However, the fate of the LNP-mRNA after endosome-engulfing and escape from the autophagy-lysosomal pathway remains unclear. It is estimated that less than 2% of the siRNA administered via LNPs escapes the endosomes.

We hypothesised that part of the LNP materials, *i.e.*, hEPO-mRNA and ionizable lipids which are engulfed in endosomes, can be incorporated into intraluminal vesicles of late-endosomes, and when these luminal vesicles of the endosomes are secreted to the extracellular milieu, part of the endocytosed material of LNPs (mRNA and ionizable lipids) will be present in EVs.

In this work, we have shown that part of the hEPO mRNA and the ionizable lipids of LNPs, which had not been dissociated/escaped into cytoplasm and had not been degraded in early endosomes through autophagy were packed into EVs (in this work these EVs are termed "endo-EVs"). The experiment was performed 13 times (seven times with MC3-LNPs and six times with DD-LNPs), and the results consistently showed that EVs contained one mole of ionizable lipids per mole of hEPO mRNA nucleotides. This differed from LNPs (both MC3- and DD-LNPs), which contained 3 times more ionizable lipids per mRNA molecule (**Fig. 1 F-J**). Despite variation in the levels of EPO-mRNA and ionizable lipids in EVs, the molar ratio between them remained the same (1:1).

In the revised manuscript, we have performed new experiments and carefully characterized these EVs in several aspects (as per guidelines by international society for extracellular vesicles: ISEV). For example we have shown that *(a)* the mean \pm SEM of endo-EVs mode size was 116.4 ± 9.23 nm in diameter determined by Nanoparticle Tracking analysis (**Fig. 2A & Suppl. Fig. S2C and D**), *(b)* the exogenous mRNA (hEPO mRNA) is present indeed inside the EVs (the mRNA is not un-specifically connected to the outside of the EV's membrane), validated by RNase treatment (**Fig. 2B and Suppl. Fig. S2E**). This characterization of EVs was further validated with additional experiments e.g. *(c)* approximately 88% of the endo-

EVs are positive for the EV markers CD63 and CD9 (**Fig. 2C**), and (**d**) approximately 26% of the CD63 and CD9 positive EVs contain the exogenous mRNA derived from endocytosis or LNPs (**Fig. 2C**).

Since these EVs were loaded with an additional/ exogenous RNA (mRNA encoding human erythropoietin (hEPO)), we investigated whether these EVs could deliver the exogenous mRNA to other cells *in vitro* and *in vivo*. The study showed for the first time that these EVs could protect the exogenous mRNA (mRNA encoding human erythropoietin, hEPO) during *in vivo* delivery to produce human EPO protein detected in plasma and organs of mice (**Fig. 3 & 4**).

Moreover, we compared, for the first time, the efficacy of LNPs and EVs for the delivery of functional mRNA to eight different organs and peripheral blood (**Fig. 3-5 and Suppl. Fig. S8 and S9**). Since these EVs contained a 3-fold lower level of ionizable lipids, compared to LNPs, we expected that the EVs would potentially cause lower expression of inflammatory cytokines in mice. The study showed that the endo-EVs cause lower expression of different inflammatory cytokines in mouse blood serum than LNPs, upon transferring equal doses of hEPO mRNA to mice (**Fig. 6**).

Moreover, we propose a hypothetical mechanism explaining the fate of LNPs inside endosomes; e.g. how part of the LNP-mRNA could be translocated to the cytoplasm or loaded (intracellularly) into the luminal vesicles of multivesicular endosomes and secreted from the cell via EVs (**Fig. 7A**). This is consistent with two independent experimental observations. First, we showed that the molar ratio of cationic ionizable lipid to mRNA nucleotide in EVs was 1:1, whereas that inside LNPs was 3:1. Second, the mRNA molecules are released from LNPs at high pH (**Fig. 7B**). This means that LNPs release their mRNA content when the pH of endosomes is raised. The mRNA that are released from LNPs, can then exit from the endosome by passive transport (mRNA neutrally charged by cationic ionisable lipid (1: 1 molar ratio) can cross the endosomal membrane and translocated into the cytoplasm, and give rise to the translation of a protein. These observations may help to address the challenge of endosomal escape of LNP-delivered mRNA, which has remained a bottleneck in the field for years.

We believe that the results presented in this manuscript, could have a significant impact on accelerating mRNA-based therapeutics and the production of safe and efficient RNA vehicles, which should not evoke immune responses in host.

Point-by-point response to reviewer comments:

Reviewer #2:

“I have re-read the paper and have also considered the arguments made by reviewer 1. In general, I do not believe the authors have responded effectively enough to Reviewer 1's critiques or my own to justify publication. I still share Reviewer 1's concerns about how they are isolating the EVs”.

Our response:

In our revised version, we have performed new experiments and characterized these EVs much more detailed than the ‘International Society for Extracellular Vesicles (ISEV) recommends (please see PMID: 30637094).

Isolation of EVs using ultracentrifugation: We apologize for any misunderstanding. Ultracentrifugation is still the gold standard method for isolating EVs when it comes to process larger sample volumes which otherwise kits cannot handle several liters of media. Several studies, including references from 36 to 46 in our manuscript and those published in Nature Communications) show Ultracentrifugation based isolation of EVs contain and protect exogenous RNA (e.g. CRISPR Cas9 mRNA, and siRNA: Nat Commun. 2018; 9(1):2359. doi: 10.1038/s41467-018-04791-8., Nature. 2015; 527(7578): 329–335. Nat Cell Biol. 2015; 17(6): 816–826. Nat Biotechnol. 2011 (4):341-5. doi: 10.1038/nbt.1807. Mol Ther. 2017; 25(7): 1580–1587. PNAS. 2017; 114(43): E9066–E9075.

In addition, we found that the method was more suitable for this study because LNPs have lower density than water (e.g. PMID: 29588418); suggesting that during ultracentrifugation, the LNPs – with lower density than water - end up at the top of the separation. EVs, which have higher density, sedimented at the bottom of a tube as a pellet (**Suppl. Fig. S3**). This also responds to reviewer #1 by answering that during ultracentrifugation LNPs according to their density cannot be pelleted with EVs.

Regarding the characterization of EVs: By ultracentrifugation, EVs were isolated from cells that had been treated with LNPs. Using immunoprecipitation (the EVs were captured using CD63 antibody), followed by flow cytometry analysis, we have shown that the same pellets that were isolated by ultracentrifuge were positive for CD63 and CD9 EV markers, which confirms that these pellets are EVs, and further confirmed by flow cytometry analysis that these EVs also contain LNP-mRNA (**Fig. 2C**). Approximately 98% of these EVs are positive for the exosome markers CD63 and CD9, and approximately 26% of the total EVs, contains mRNA that is derived from endocytosis of LNPs (**Fig. 2C**).

Moreover, the reviewer #1 suggested that EVs isolated from LNP-administered cells (endo-EVs), may need additional characterization.

Additional characterization of endo-EVs:

We performed additional characterization. Isolated EVs were treated with RNase and the result showed that the hEPO mRNA - EVs obtained from endocytosis of LNPs - is indeed inside of these EVs (the mRNA is not un-specifically connected to the outside of the EV's membrane). No significant difference could be detected between RNase treated and non-RNase treated EVs (**Fig. 2B and Suppl. Fig. S2E**). (*This is common practice in EV field and we remain thankful to Reviewers who suggested this experiment, based on which we validated the presence of mRNA inside EVs*). Nanoparticle tracking analysis (NTA) is another way to characterize EVs that is for the determination of size and concentration of EVs. Based on comments from reviewer#1 we performed NTA and showed that the size of EVs is in the range of what EVs have been shown (**Fig. 2A and Suppl. Fig. S2C and D**). We could show that (i) endo-EVs have an average size of 162.5 nm in diameter, which is within the size of the EVs, including exosomes.

The results of these new experiments have been included in the revised version of the manuscript (**Fig. 2A-C and Suppl. Fig. S2C-E and page 8-10 of the manuscript_clean version**).

We hope that the results and changes that we have included in the manuscript are sufficient and convince Reviewer #2 regarding remarks about the “isolation” and “characterization” of EVs.

Reviewer #2:

“Rebuttal to Reviewer 1, point 2. In Figure 2, the authors write that they show **delivery to B cells**. It is unclear if they do. If they are quantifying GFP, then they do. If (more likely) they are reading out Cy5, they don't... they only have uptake in the cells. Moreover, this is not the first time B cells have been targeted”.

Our response:

In fact, initially, reviewer #1 suggested to remove figure 2. We could have removed the figure 2, but the editor kindly suggested us to keep the figure 2. **In the revised manuscript, we have moved these results to "Suppl. Fig. S5"** (merged with other in vitro studies), **and in addition we describe clearly that these-EVs can deliver fluorescence labelled mRNA (Cy5 mRNA) to B cell, T cells, and monocytes in vitro.**

We deliver Cy5- mRNA to cells via EVs, and measure the uptake of Cy5- mRNA by the cells. We quantified the percentage of cells taken up the Cy5 mRNA, e.g. 70% of the HTB-177 cells were positive for (uptake) Cy5- mRNA, and 6% of B-cells which uptake the Cy5-

mRNA (few examples to refer). For details, please refer to text of **Suppl. Fig. S5 A-D** on page #10 of the revised version of manuscript, and legends text of **Suppl. Fig. S5 A-D**.

However, we agree with the reviewer#2 that this is not the first time B cells have been targeted. We do not claim that B cells are transfected for the first time, but we claim that it is the first study to show delivery of exogenous mRNA (Cy5 mRNA) via EVs. Indeed, it does not solve the problems of targeting B cells, but opens up a new opportunity that EVs can be alternative vehicles to transfer mRNA to B-cells in future. How successful it will be, will indeed require further studies.

Reviewers #2:

“Rebuttal to me, point (4). I still do not see how - from a conservation of mass perspective - their results can be explained. I understand that the potential efficacy will differ, but it still seems strange that more RNA can go into the cell AND more RNA can go out of the cell at the same time”.

(4) It is hard to understand how the MC3 LNPs could be simultaneously delivering more mRNA into the cytoplasm (leading to more delivery in Figure 1B, C) AND also be recycled out more in the EVs (Fig. 1F).

Our response:

We are sorry because we think there has been a misunderstanding. Let us explain in detail. The total mRNA delivered to cells is 100 μg , out of which approximately 76 fg (femto-gram) could be detected in cells after 4 days (**Fig. 1A**) and about 2 μg could be detected in total EVs secreted by cells after 4 days (**Fig. 1F**), which in total is not equal to 100 μg of mRNA delivered to cells. When we measured the mRNA in the cells (cytoplasm), this did not consider the mRNA amount which is already secreted in the EVs.

In addition, our results (**Fig. 1, Suppl. Fig. S1, Suppl. Fig.S8, and Suppl. Fig. S9**) show that more hEPO mRNA is translated into protein (indicating that more hEPO mRNA does endosomal escape) when MC3-LNPs deliver mRNA to cells compared to DD-LNPs.

We hope this explanation is the answer to the Reviewer's question.

Reviewers #2:

“If the paper is published, I think the authors should remove all statements speculating these vehicles > LNPs. In their rebuttal, they say to reviewer 1 (and to me, separately) that it's too early to compare EVs to LNPs. However, in their abstract they write 'Compared with LNPs, EVs elicit significantly less inflammatory response, and may function as a safer vehicle for exogenous mRNA delivery.' That (big big big) statement is not substantiated by (1) their own rebuttal or (2) their data”.

Our response:

In the revised manuscript, we have introduced the necessary changes to the manuscript texts.

For example, in **Fig. 6**, and in any places of our manuscript as well as in the response letter, we remained consistent to emphasize that LNPs are most advanced delivery vehicles, and are already in clinical trials. However, we claim that this is first study to show the delivery of mRNA via EVs, and since EVs are natural biological carriers, and contain 1/3 less ionizable lipid than LNPs, the EVs could cause lower expression of proinflammatory cytokines. Regarding the statement “EVs elicit significantly less inflammatory response” as a claim; here, we agree with the reviewer. In the modified version, we have changed our expression that these EVs cause lower expression of different inflammatory cytokines in mouse blood serum than LNPs (please see abstract, results and discussion).

We hope these changes are sufficient measures. However, if there are remaining concerns, we would be happy to make further corrections if required.

Changes introduced in the revised manuscript attached:

Changes included to the revised manuscript text: for the detail changes please see the attached file "Revised_Manuscript Tracked_Maugeri et al"

Changes included to the manuscript figures with additional data:

- **New “Fig. 2”** (*Figure main manuscript*)
- **Modified “Suppl. Fig. S1”** (*Figure supplementary material*)
- **Modified “Suppl. Fig. S2”** (*Figure supplementary material*)
- **Suppl. Fig. S4**
- **Modified “Suppl. Fig. S5”** (*Figure supplementary material*)
- **Changes included to the manuscript text:** (*Figures description text, supplementary figures text and supplementary material*)
- **Text for results** (main manuscript, results section)
- **Text for experiments** (materials and methods section)
- **Source data files:** (Source data files updated)

Kind regards,

Hadi Valadi

REVIEWERS' COMMENTS:

Reviewer #2 (Remarks to the Author):

The authors have addressed all of my concerns.